# SOLVING ROBUST MDPS
# AS A SEQUENCE OF STATIC RL PROBLEMS

## ABSTRACT

Designing control policies whose performance level is guaranteed to remain above a given threshold in a span of environments is a critical feature for the adoption of reinforcement learning (RL) in real-world applications. The search for such robust policies is a notoriously difficult problem, related to the so-called dynamic model of transition function uncertainty, where the environment dynamics are allowed to change at each time step. But in practical cases, one is rather interested in robustness to a span of static transition models throughout interaction episodes. The static model is known to be harder to solve than the dynamic one, and seminal algorithms, such as robust value iteration, as well as most recent works on deep robust RL, build upon the dynamic model. In this work, we propose to revisit the static model. We suggest an analysis of why solving the static model under some mild hypotheses is a reasonable endeavor, based on an equivalence with the dynamic model, and formalize the general intuition that robust MDPs can be solved by tackling a series of static problems. We introduce a generic meta-algorithm called IWOCS, which incrementally identifies worst-case transition models so as to guide the search for a robust policy. Discussion on IWOCS sheds light on new ways to decouple policy optimization and adversarial transition functions and opens new perspectives for analysis. We derive a deep RL version of IWOCS and demonstrate it is competitive with state-of-the-art algorithms on classical benchmarks.

## 1 INTRODUCTION

One major obstacle in the way of real-life deployment of reinforcement learning (RL) algorithms is their inability to produce policies that retain, without further training, a guaranteed level of efficiency when controlling a system that somehow differs from the one they were trained upon. This property is referred to as *robustness*, by opposition to *resilience*, which is the ability to recover, through continued learning, from environmental changes. For example, when learning control policies for aircraft stabilization using a simulator, it is crucial that the learned controller be able to control a span of aircraft configurations with different geometries, or masses, or in various atmospheric conditions. Depending on the criticality of the considered application, one will prefer to optimize the expected performance over a set of environments (thus weighting in the probability of occurrence of a given configuration) or, at the extreme, optimize for the worst case configuration. Here, we consider such worst case guarantees and revisit the framework of robust Markov Decision Processes (MDPs) (Iyengar, 2005).

Departing from the common perspective which views robust MDPs as two-player games, we investigate whether it is possible to solve them through a series of non-robust problems. The two-player game formulation is called the dynamic model of transition function uncertainty, as an adversarial environment is allowed to change the transition dynamics at each time step. The solution to this game can be shown to be equivalent, for stationary policies and rectangular uncertainty sets, to that of the static model, where the environment retains the same transition function throughout the time steps.

Our first contribution is a series of arguments which cast the search for a robust policy as a resolution of the static model (Section 2). We put this formulation in perspective of recent related works in robust RL (Section 3). Then, we introduce a generic meta-algorithm which we call IWOCS for Incremental Worst-Case Search (Section 4). IWOCS builds upon the idea of incrementally identifying worst case transition functions and expanding a discrete uncertainty set, for which a robust policy

can be approximated through a finite set of non-robust value functions. We instantiate two IWOCS algorithms, one on a toy illustrative problem with a discrete state space, then another on popular, continuous states and actions, robust RL benchmarks where it is shown to be competitive with state-of-the art robust deep RL algorithms (Section 5).

## 2  PROBLEM STATEMENT

**Reinforcement Learning**   (RL) (Sutton & Barto, 2018) considers the problem of learning a decision making policy for an agent interacting over multiple time steps with a dynamic environment. At each time step, the agent and environment are described through a state $s \in \mathcal{S}$, and an action $a \in \mathcal{A}$ is performed; then the system transitions to a new state $s'$ according to probability $T(s'|s, a)$, while receiving reward $r(s, a, s')$. The tuple $M_T = (\mathcal{S}, \mathcal{A}, T, r)$ forms a Markov Decision Process (MDP) (Puterman, 2014), which is often complemented with the knowledge of an initial state distribution $p_0(s)$. Without loss of generality and for the sake of readability, we will consider a unique starting state $s_0$ in this paper, but our results extend straightforwardly to a distribution $p_0(s)$. A stationary decision making policy is a function $\pi(a|s)$ mapping states to distributions over actions (writing $\pi(s)$ the action for the special case of deterministic policies). Training a reinforcement learning agent in MDP $M_T$ consists in finding a policy that maximizes the expected $\gamma$-discounted return from $s_0$: $J_T^\pi = \mathbb{E}[\sum_{t=0}^{\infty} \gamma^t r(s_t, a_t, s_{t+1})|s_0, a_t \sim \pi, s_{t+1} \sim T] = V_T^\pi(s_0)$, where $V_T^\pi$ is the value function of $\pi$ in MDP $M_T$, and $\gamma \in [0, 1)$. An optimal policy in $M_T$ will be noted $\pi_T^*$ and its value function $V_T^*$. A convenient notation is the state-action value function $Q_T^\pi(s, a) = \mathbb{E}_{s' \sim T}[r(s, a, s') + \gamma V_T^\pi(s')]$ of policy $\pi$ in MDP $M_T$, and the corresponding optimal $Q_T^*$. Key notations are summarized in Appendix C.

**Robust MDPs,**   as introduced by Iyengar (2005) or Nilim & El Ghaoui (2005), introduce an additional challenge. The transition functions $T$ are picked from an uncertainty set $\mathcal{T}$ and are allowed to change at each time step, yielding a sequence $\mathbf{T} = \{T_t\}_{t \in \mathbb{N}}$. A common assumption, called *sa-rectangularity*, states that $\mathcal{T}$ is a Cartesian product of independent marginal sets of distributions on $\mathcal{S}$, for each state-action pair. The value of a stationary policy $\pi$ in the sequence of MDPs induced by $\mathbf{T} = \{T_t\}_{t \in \mathbb{N}}$ is noted $V_{\mathbf{T}}^\pi$. The pessimistic value function for $\pi$ is $V_{\mathcal{T}}^\pi(s) = \min_{\mathbf{T}} V_{\mathbf{T}}^\pi(s)$, where the agent plays a sequence of actions $a_t \in \mathcal{A}$ drawn from $\pi$, against the environment, which in turn picks transition models $T_t \in \mathcal{T}$ so as to minimize the overall return. The robust value function is the largest such pessimistic value function and hence the solution to $V_{\mathcal{T}}^*(s) = \max_\pi \min_{\mathbf{T}} V_{\mathbf{T}}^\pi(s)$. The robust MDP problem can be cast as the zero-sum two-player game, where $\hat{\pi}$ denote the decision making policy of the adversarial environment, deciding $T_t \in \mathcal{T}$ based on previous observations. Then, the problem becomes $\max_\pi \min_{\hat{\pi}} V_{\hat{\pi}}^\pi(s)$, where $V_{\hat{\pi}}^\pi$ is the expected value of a trajectory where policies $\pi$ and $\hat{\pi}$ play against each other. Hence, the optimal policy becomes the minimax policy, which makes it robust to all possible future evolutions of the environment's properties.

**Robust Value Iteration.**   Following Iyengar (2005, Theorem 3.2), the optimal robust value function $V_{\mathcal{T}}^*(s) = \max_\pi \min_{\mathbf{T}} V_{\mathbf{T}}^\pi(s)$ is the unique solution to the robust Bellman equation $V(s) = \max_a \min_T \mathbb{E}_{s' \sim T}[r(s, a, s') + \gamma V(s')] = \mathcal{L}V(s)$. This directly translates into a robust value iteration algorithm which constructs the $V_{n+1} = \mathcal{L}V_n$ sequence of value functions (Satia & Lave Jr, 1973; Iyengar, 2005). Such robust policies are, by design, very conservative, in particular when the uncertainty set is large and under the rectangularity assumption. Several attempts at mitigating this intrinsic over-conservativeness have been made from various perspectives. For instance, Lim et al. (2013) propose to learn and tighten the uncertainty set, echoing other works that incorporate knowledge about this set into the minimax resolution (Xu & Mannor, 2010; Mannor et al., 2012). Other approaches (Wiesemann et al., 2013; Lecarpentier & Rachelson, 2019; Goyal & Grand-Clement, 2022) propose to lift the rectangularity assumption and capture correlations in uncertainties across states or time steps, yielding significantly less conservative policies. Ho et al. (2018) and Grand-Clément & Kroer (2021) retain the rectangularity assumption and propose algorithmic schemes to tackle large but discrete state and action spaces.

**The static model.**   In many applications, one does not wish to consider non-stationary transition functions, but rather to be robust to any transition function from $\mathcal{T}$ which remains stationary throughout a trajectory. This is called the *static* model of transition function uncertainty, by opposition to the *dynamic* model where transition functions can change at each time step. Hence, the static model's minimax game boils down to $\max_\pi \min_T V_T^\pi(s)$. If the agent is restricted to stationary

policies $\pi(a|s)$, then $\max_\pi \min_{\mathbf{T}} V_{\mathbf{T}}^\pi(s) = \max_\pi \min_T V_T^\pi(s)$ (Iyengar, 2005, Lemma 3.3), that is the static and dynamic problems are equivalent, and the solution to the dynamic problem is found for a static adversary.[1] In this paper, we will only consider stationary policies.

**No-duality gap.** Wiesemann et al. (2013, Equation 4 and Proposition 9) introduce an important saddle point condition stating that $\max_\pi \min_T V_T^\pi(s) = \min_T \max_\pi V_T^\pi(s)$. [2]

**Incrementally solving the static model.** Combining the static and dynamic models equivalence and the no-duality gap condition, we obtain that, for rectangular uncertainty sets and stationary policies, the optimal robust value function $V_{\mathcal{T}}^*(s) = \max_\pi \min_{\mathbf{T}} V_{\mathbf{T}}^\pi(s) = \max_\pi \min_T V_T^\pi(s) = \min_T \max_\pi V_T^\pi(s) = \min_T V_T^*(s)$. The key idea we develop in this paper stems from this formulation. Suppose we are presented with $M_{T_0}$ and solve it to optimality, finding $V_0^*(s) = V_{T_0}^*(s)$. Then, suppose we identify $M_{T_1}$ as a possible better estimate of a worst case MDP in $\mathcal{T}$ than $T_0$. We can solve for $V_{T_1}^*$ and $V_1^*(s) = \min\{V_{T_0}^*(s), V_{T_1}^*(s)\}$ is the robust value function for the discrete uncertainty set $\mathcal{T}_1 = \{T_0, T_1\}$. The intuition we attempt to capture is that by incrementally identifying candidate worst case MDPs, one should be able to define a sequence of discrete uncertainty sets $\mathcal{T}_i = \{T_j\}_{j \in [0,i]}$ whose robust value function $V_i^*$ decreases monotonously, and may converge to $V^*$. In other words, it should be possible to incrementally robustify a policy by identifying the appropriate sequence of transition models and solving individually for them, trading the complexity of the dynamic model's resolution for a sequence of classical MDP problems. The algorithm we propose in Section 4 follows this idea and searches for robust stationary policies for the dynamic model, using the static model, by incrementally growing a finite uncertainty set.

# 3 RELATED WORK

**Robust RL as two-player games.** A common approach to solving robust RL problems is to cast the dynamic formulation as a zero-sum two player game, as formalized by Morimoto & Doya (2005). In this framework, an adversary, denoted by $\hat{\pi} : \mathcal{S} \to \mathcal{T}$, is introduced, and the game is formulated as $\max_\pi \min_{\hat{\pi}} \mathbb{E}[\sum_{t=0}^\infty \gamma^t r(s_t, a_t, s_{t+1})|s_0, a_t \sim \pi(\cdot|s_t), T_t = \hat{\pi}(s_t, a_t), s_{t+1} \sim T_t(\cdot|s_t, a_t)]$. Most methods differ in how they constrain $\hat{\pi}$'s action space within the uncertainty set. A first family of methods define $\hat{\pi}(s_t) = T_{ref} + \Delta(s_t)$, where $T_{ref}$ denotes the reference transition function. Among this family, Robust Adversarial Reinforcement Learning (RARL) (Pinto et al., 2017) applies external forces at each time step $t$ to disturb the reference dynamics. For instance, the agent controls a planar monopod robot, while the adversary applies a 2D force on the foot. In noisy action robust MDPs (NR-MDP) (Tessler et al., 2019) the adversary shares the same action space as the agent and disturbs the agent's action $\pi(s)$. Such gradient-based approaches incur the risk of finding stationary points for $\pi$ and $\hat{\pi}$ which do not correspond to saddle points of the robust MDP problem. To prevent this, Mixed-NE (Kamalaruban et al., 2020) defines mixed strategies and uses stochastic gradient Langevin dynamics. Similarly, Robustness via Adversary Populations (RAP) (Vinitsky et al., 2020) introduces a population of adversaries, compelling the agent to exhibit robustness against a diverse range of potential perturbations rather than a single one, which also helps prevent finding stationary points that are not saddle points. Aside from this first family, State Adversarial MDPs (Zhang et al., 2020; 2021; Stanton et al., 2021) involve adversarial attacks on state observations, which implicitly define a partially observable MDP. The goal in this case is not to address robustness to the worst-case transition function but rather against noisy, adversarial observations. A third family of methods considers the general case of $\hat{\pi}(s_t) = T_t$ where $T_t \in \mathcal{T}$. Minimax Multi-Agent Deep Deterministic Policy Gradient (M3DDPG) (Li et al., 2019) is designed to enhance robustness in multi-agent reinforcement learning settings, but boils down to standard robust RL in the two-agents case. Max-min TD3 (M2TD3) (Tanabe et al., 2022) considers a policy $\pi$, defines a value function $Q(s, a, T)$ which approximates $Q_T^\pi(s, a) = \mathbb{E}_{s' \sim T}[r(s, a, s') + \gamma V_T^\pi(s')]$, updates an adversary $\hat{\pi}$ so as to minimize $Q(s, \pi(s), \hat{\pi}(s))$ by taking a gradient step with respect to $\hat{\pi}$'s parameters, and updates the policy $\pi$ using a TD3 gradient update in the direction maximizing $Q(s, \pi(s), \hat{\pi}(s))$. As such, M2TD3 remains a robust value iteration method which solves the dynamic problem by alternating updates on $\pi$ and $\hat{\pi}$, but since it approximates $Q_T^\pi$, it is also closely related to the method we introduce

---

[1]This does not imply the solution to the static model is the same as that of the dynamic model in general: the optimal static $\pi$ may be non-stationary and finding it is known to be NP-hard.

[2]The static-dynamic equivalence and the no-duality gap property's context is recalled in Appendix B.

in the next section. Wang et al. (2023) introduced a policy gradient method for robust MDPs with global convergence guarantees. While their work shares some conceptual similarities with ours in optimizing policies using a static model, it differs in key aspects. Their approach is limited to policy-based methods, whereas ours is more versatile, applicable to any RL algorithm, and scalable to larger state and action spaces.

**Regularization.** Derman et al. (2021); Eysenbach & Levine (2022) also highlighted the strong link between robust MDPs and regularized MDPs, showing that a regularized policy learned during interaction with a given MDP was actually robust to an uncertainty set around this MDP. Kumar et al. (2023) propose a promising approach in which they derive the adversarial transition function in a closed form and demonstrate that it is a rank-one perturbation of the reference transition function. This simplification results in more streamlined computation for the robust policy gradient.

**Domain randomization** (DR) (Tobin et al., 2017) learns a value function $V(s) = \max_\pi \mathbb{E}_{T \sim \mathcal{U}(\mathcal{T})} V_T^\pi(s)$ which maximizes the expected return *on average* across a fixed distribution on $\mathcal{T}$. As such, DR approaches do not optimize the worst-case performance. Nonetheless, DR has been used convincingly in applications (Mehta et al., 2020; OpenAI et al., 2019). Similar approaches also aim to refine a base DR policy for application to a sequence of real-world cases (Lin et al., 2020; Dennis et al., 2020; Yu et al., 2018).

For a more complete survey of recent works in robust RL, we refer the reader to the work of Moos et al. (2022). To the best of our knowledge, the approach sketched in the previous section and developed in the next one is the only one that directly addresses the static model. For that purpose, it exploits the equivalence with the dynamic model for stationary policies and solves the dual of the minimax problem, owing to the no-duality gap property.

## 4 Incremental Worst-case Search

In order to search for robust policies, we consider the no-duality gap property: the best performance one can expect in the face of transition function uncertainty $\max_\pi \min_T V_T^\pi(s_0)$, is also the worst performance the environment can induce for each transition function's optimal policy $\min_T V_T^*(s_0)$. If the value $V_T^\pi(s_0)$ was strictly concave/convex with respect to $\pi/T$ respectively, we could hope to solve for the robust policy through a (sub)gradient ascent/descent method. Unfortunately, it seems $V_T^\pi(s_0)$ easily admits more convoluted optimization landscapes, involving stationary points, local minima and maxima. The $\max_\pi$ problem often benefits from regularization (Geist et al., 2019). Although one could study regularization for the $\min_T$ problem (Grand-Clément & Petrik, 2022) or the equivalence with a regularized objective (Derman et al., 2021), we turn towards a simpler process conceptually.

**Algorithm.** We consider a (small) discrete set of MDPs $\mathcal{T}_i = \{T_j\}_{j \in [0,i]}$, for which we derive the corresponding optimal value functions $Q_{T_j}^*$. Then we define $Q_i$ as the function that maps any pair $s, a$ to the smallest expected optimal outcome $Q_i(s, a) = \min_{j \in [0,i]} \{Q_{T_j}^*(s, a)\}$. The corresponding greedy policy is $\pi_i(s) \in \arg\max_a Q_i(s, a)$ and is a candidate for the robust policy. Let us define $T_{i+1} \in \arg\min_{T \in \mathcal{T}} V_T^{\pi_i}(s_0)$. Then, if $V_{T_{i+1}}^{\pi_i}(s_0) = Q_i(s_0, \pi_i(s_0))$, we have found a robust policy for all transition models in $\mathcal{T}$. Otherwise, we can solve for $Q_{T_{i+1}}^*$, append $T_{i+1}$ to $\mathcal{T}_i$ to form $\mathcal{T}_{i+1}$, and repeat. Consequently, the idea we develop is to incrementally expand $\mathcal{T}_i$ by solving $\min_{T \in \mathcal{T}} V_T^{\pi_i}(s_0)$ using optimization methods that can cope with ill-conditioned optimization landscapes. We call Incremental Worst Case Search (IWOCS) this general method, which we summarize in Algorithm 1.

**Rectangularity.** One should note that $\mathcal{T}_i$ is a subset of a (supposed) $sa$-rectangular uncertainty set, but is not $sa$-rectangular itself, so there is no guarantee that the static-dynamic equivalence holds in $\mathcal{T}_i$, and $Q_i$ is a pessimistic value function for the static case only, on the $\mathcal{T}_i$ uncertainty set. However, one can consider the $sa$-rectangular set $\tilde{\mathcal{T}}_i = \times_{s,a} \{T_j(\cdot|s, a)\}_{j \in [0,i]}$ composed of the cartesian product of all local $\{T_j(\cdot|s, a)\}_{j \in [0,i]}$ sets for each $s, a$ pair.

**Property 1.** *For any $i, s, a$, we have*

$$Q_i(s, a) \geqslant Q_{\tilde{\mathcal{T}}_i}^*(s, a) \geqslant Q_{\mathcal{T}}^*(s, a).$$

---

**Algorithm 1** Incremental Worst-Case Search meta-algorithm (in blue: the sub-algorithms)

---

**Input:** $\mathcal{T}, T_0$, max nb of iterations $M$, tolerance on robust value $\epsilon$
**for** $i = 0$ to $M$ **do**
  Find non-robust $Q_{T_i}^* = \max_\pi Q_{T_i}^\pi$
  Define $\mathcal{T}_i = \{T_j\}_{j \leqslant i}$
  Define robust $Q_i : s, a \mapsto \min_{j \leqslant i} \{Q_{T_j}^*(s, a)\}$
  Define candidate $\pi_i(s) = \arg\max_a(Q_i(s, a))$
  Find worst $T_{i+1} = \arg\min_{T \in \mathcal{T}} V_T^{\pi_i}(s_0)$
  **if** $|V_{T_{i+1}}^{\pi_i}(s_0) - Q_i(s_0, \pi_i(s_0))| \leqslant \epsilon$ **then**
    **return** $\pi_i, T_{i+1}, V_{T_{i+1}}^{\pi_i}(s_0)$
  **end if**
**end for**
**return** $\pi_M, T_{M+1}, V_{T_{M+1}}^{\pi_M}(s_0)$

---

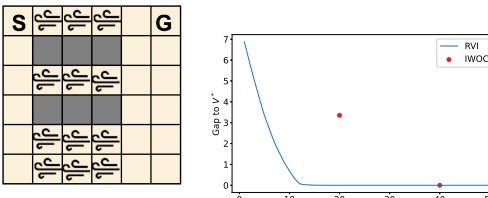

Figure 1: Convergence to $V^*$ vs Bellman iterates (right) in the Windy walk grid-world (left).

The proof follows directly from the fact that $\mathcal{T}_i \subset \tilde{\mathcal{T}}_i \subset \mathcal{T}$. We abusively call $Q_i$ the $\mathcal{T}_i$-robust value function. In $s_0$, $Q_i$ coincides with the robust value function for the static model of uncertainty with respect to the $\mathcal{T}_i$ uncertainty set.

**Property 2.** *For any $i, s, a$, we have*

$$Q_{i+1}(s, a) \leqslant Q_i(s, a).$$

The proof is immediate as well since $Q_{i+1}$ drawn among the same finite set of functions as $Q_i$, complemented with $Q_{T_{i+1}}^*$. Hence the $Q_i$ functions form a monotonically decreasing sequence. Since $Q_i$ is lower bounded (by $Q_\mathcal{T}^*$), IWOCS is necessarily convergent.[3]

**Choosing $T_{i+1}$.** One could define a variant of Algorithm 1 which picks $T_{i+1}$ using another criterion than the worst-case transition model for $\pi_i$, for instance by drawing $T_{i+1}$ uniformly at random, without loosing the two properties above. This underlines that the procedure for choosing $T_{i+1}$ is a heuristic part of IWOCS. In all cases, the sequence of $Q_i$ remains monotonous and hence convergence in the limit remains guaranteed. Specifically, if, in the limit, $\mathcal{T}_i$ converges to $\mathcal{T}$ (under some appropriate measure on uncertainty sets), then $Q_i$ converges to the robust value function by definition. Whether this occurs or not, strongly depends on how $T_{i+1}$ is chosen at each iteration. In particular, premature stopping can occur if $T_{i+1}$ is among $\mathcal{T}_i$. We conjecture choosing the worst-case transition model for $\pi_i$ is an intuitive choice here, and reserve further theoretical analysis on this matter for future work. One bottleneck difficulty of this selection procedure for $T_{i+1}$ lies in solving the $\min_T$ problem accurately enough. However this difficulty is decoupled from that of the policy optimization process, which is only concerned with static MDPs.

**Illustration.** We implement an IWOCS algorithm on a toy example, using value iteration (VI) as the policy optimization algorithm and a brute force search across transition functions to identify worst-case MDPs ($V_T^{\pi_i}(s_0)$ is evaluated through Monte-Carlo rollouts). Detailed pseudo-code is provided in Appendix E. The goal here is to illustrate the behavior of IWOCS, compare it to the seminal robust value iteration (RVI) algorithm, and validate empirically that IWOCS is able to find worst-case static MDPs and robust policies. This vanilla IWOCS is evaluated on the "windy walk" grid-world

---

[3]Although not necessarily to $Q_\mathcal{T}^*$.

MDP illustrated on Figure 1, where an agent wishes to navigate from a starting position $S$ to a goal $G$. Actions belong to the discrete $\{N, S, E, W\}$ set and transitions are deterministic, except in the "corridors", where wind can knock back the agent to the left. In the topmost corridor, the probability of being knocked left is $\alpha$, in the middle corridor it is $\alpha^3$ and it is $\alpha^6$ in the bottom corridor. Hence, the uncertainty set is fully parameterized by $\alpha$, which takes 25 discrete values, uniformly distributed in $[0, 0.5]$. Rewards are $-1$ at each time step and the goal is an absorbing state yielding zero reward.

Figure 1 illustrates how IWOCS converges to the robust value function $V^*$. RVI builds the sequence $V_{n+1} = \mathcal{L}V_n$ and we plot $|V_n(s_0) - V_{\mathcal{T}}^*(s_0)|$ versus the number of robust Bellman iterates $n$. On the other hand, IWOCS requires its first policy optimization to terminate before it can report its first $Q_i(s_0, \pi_i(s_0))$. Thus, we plot $|Q_i(s_0, \pi_i(s_0)) - V_{\mathcal{T}}^*(s_0)|$ after a fixed number of 100 standard Bellman backups for VI. It is important to note that one iterate of the standard Bellman operator requires solving a $\max_a$ in each state, while an iterate of the robust Bellman operator requires solving a more costly $\max_a \min_T$ problem in each state. Therefore, the x-axis does not account for computational time. IWOCS finds the worst-case static model after two iterations and converges to the same value as RVI.

**Computational complexity.** Recall that the complexity of robust value iteration (RVI), in discrete state and action MDPs, and for a $sa$-rectangular uncertainty set, is $O(cn_S^2 n_A \log(1/\epsilon)/\log(1-\gamma))$, where $n_S$ is the number of states, $n_A$ the number of actions, $\epsilon$ is the tolerance for the robust value function and $c$ is the cost of computing a single $\min_T$ solution (Iyengar, 2005). Recall also that the complexity of value iteration (VI) is $O(n_S^2 n_A \log(1/\epsilon)/\log(1-\gamma))$ (VI is a special case of RVI with a singleton as uncertainty set, so $c = 1$). Comparing IWOCS and RVI is a delicate matter because IWOCS is not based on a contraction mapping and has no convergence guarantees to the robust value function. Consequently, comparisons should be taken with a grain of salt. Yet, it is legitimate to wonder whether one can analyse the time complexity of IWOCS versus RVI. One iteration of IWOCS in discrete state and action spaces, as presented in Section 4, has the complexity of VI for the policy search part, plus the complexity of finding a worst case transition function in an $sa$-rectangular uncertainty set, which is $O(cn_S n_A)$. Hence, the overall complexity for $M$ iterations of IWOCS is $O(M(n_S^2 n_A \log(1/\epsilon)/\log(1-\gamma) + cn_S n_A))$. Compared to RVI, this bound will be smaller when $c$ is large, which is the case when one deals with complex uncertainty sets and without further hypotheses. This short discussion provides a rationale to why IWOCS might be a time-efficient algorithm in large scale robust RL problems.

## 5 DEEP IWOCS

We now turn towards challenging robust control problems and introduce an instance of IWOCS meant to accommodate large and continuous state and action spaces, using function approximators such as neural networks. This instance of IWOCS uses Soft Actor Critic (SAC) Haarnoja et al. (2018) as the policy optimization method, as it has been proven to yield a locally robust policy around the MDP it is trained upon (Eysenbach & Levine, 2022). Our code is available at https://anonymous.url and experimental computing setup is summarized in Appendix D.

### 5.1 METHOD

**Accounting for regularization terms.** Since SAC learns a regularized Q-function which accounts for the policy's entropy, and lets the importance of this term vary along the optimization, orders of magnitude may change between $Q_{T_i}$ and $Q_{T_j}$. To avoid the influence of this regularization term when defining the $\mathcal{T}_i$-robust Q-function, we train an additional unregularized Q-network which only takes rewards into account. We call $\pi_T$ the policy network which approximates the optimal policy of the regularized MDP based on $T$. This policy's (regularized) value function is approximated by the $Q_T'$ network (our implementation uses double Q-networks as per the common practice — all details in Appendix F), while an additional $Q_T$ network (double Q-network also) tracks the unregularized value function of $\pi_T$. The $\mathcal{T}_i$-robust Q-function is defined with respect to this unregularized value function as $Q_i(s, a) = \min_{j \in [0,i]} \{Q_{T_j}(s, a)\}$.

**Partial state space coverage.** In large state space MDPs, it is likely that interactions will not explore the full state space. Consequently, the different $Q_{T_j}$ functions are trained on replay buffers whose empirical distribution's support my vary greatly. Evaluating neural networks outside of their training

distribution is prone to generalization errors. This begs for indicator functions specifying on which $(s, a)$ pair each $Q_T$ is relevant. We chose to implement such an indicator function using predictive coding (Rao & Ballard, 1999) on the dynamical model $T_j$. Note that other choices can be equally good (or better), such as variance networks (Neklyudov et al., 2019), ensembles of neural networks (Lakshminarayanan et al., 2016) or 1-class classification (Béthune et al., 2023). Our predictive coding model for $T_j$ predicts $\hat{T}_j(s, a) = s'$ for deterministic dynamics, by minimizing the expected value of the loss $\ell(\hat{T}_j(s, a); s') = \|\hat{T}_j(s, a) - s'\|_1$. At inference time, along a trajectory, we consider $Q_{T_j}$ has been trained on sufficient data in $s_t, a_t$, if $\ell(\hat{T}_j(s_{t-1}, a_{t-1}); s_t) \leqslant \rho_j$, ie. if the prediction error for $s_t$ is below the threshold $\rho_j$ (details about tuning $\rho_j$ in Appendix G). We set $Q_{T_j}^*$ to be $+\infty$ in all states where $\ell(\hat{T}_j(s_{t-1}, a_{t-1}); s_t) > \rho_j$, so that it does not participate in the definition of $Q_i$.

**Worst case identification.** When $V_T^{\pi_i}(s_0)$ is non differentiable with respect to $T$ (or $T$'s parameters), one needs to fall back on black-box optimization to find $T_{i+1} = \arg\min_{T \in \mathcal{T}} V_T^{\pi_i}(s_0)$. We turn to evolutionary strategies, and in particular CMA-ES (Hansen & Ostermeier, 2001) for that purpose, for its ability to escape local minima and efficiently explore the uncertainty set $\mathcal{T}$ even when the latter is high-dimensional (hyperparameters in Appendix F). Note that making $V_T^{\pi}(s_0)$ differentiable with respect to $T$ is feasible by making the critic network explicitly depend on $T$'s parameters, as in the work of Tanabe et al. (2022). We do not resort to such a model, as it induces the risk for generalization errors, but it constitutes a promising alternative for research. To evaluate $V_T^{\pi_i}(s_0)$ for a given $T$, we run a roll-out from $s_0$ by applying $\pi_i(s)$ in each encountered state $s$. Since we consider continuous action spaces and keep track of the critics $Q_{T_j}$, $Q'_{T_j}$ and the actor $\pi_{T_j}$ for all $T_j \in \mathcal{T}_i$, we can make direct use of $\pi_{T_j}$ which is designed to mimic an optimal policy in $M_{T_j}$. Specifically, in $s$, we evaluate $j^* = \arg\min_{j \leqslant i} Q_{T_j}^*(s, \pi_{T_j}(s))$, and apply $\pi_i(s) = \pi_{j*}(s)$. If no $Q_{T_j}^*$ is valid in $s$, we fall back to a default policy trained with domain randomization.

## 5.2 EMPIRICAL EVALUATION

**Experimental framework.** This section assesses the proposed algorithm's worst-case performance and generalization capabilities. Experimental validation is performed on optimal control problems using the MuJoCo simulation environments[4] (Todorov et al., 2012). IWOCS is benchmarked against state-of-the-art robust reinforcement learning methods, including M2TD3 (Tanabe et al., 2022), M3DDPG (Li et al., 2019), and RARL (Pinto et al., 2017). We also compare with Domain Randomization (DR) (Tobin et al., 2017) for completeness. For each environment, two versions of the uncertainty set are considered, following the benchmarks reported by Tanabe et al. (2022). In the first one, $T$ is parameterized by a global friction coefficient and the agent's mass. In the second one, a third, environment-dependent parameter is included (details in Appendix J). To ensure a fair comparison we also aligned with the sample budget of Tanabe et al. (2022): performance metrics were collected after 4 million steps for environments with a 2D uncertainty set and after 5 million steps for those with a 3D uncertainty set. All reference methods optimize a single policy along these 4 or 5 million steps, but IWOCS optimizes a sequence of non-robust policies, for which we divide this sample budget: we constrain IWOCS to train its default policy and each subsequent SAC agent for a fixed number of interaction steps, so that the sum is 4 or 5 million steps (Appendix H and I).[5] Results for all methods other than IWOCS are taken from the work of Tanabe et al. (2022). All results reported below are averaged over 10 distinct random seeds.

**IWOCS\*.** We define a variant of IWOCS by replacing CMA-ES with a plain grid search across the uncertainty set, mimicking the worst case search of Tanabe et al. (2022), in order to assess whether CMA-ES effectively finds adequate worst-case transition models. Contrarily to CMA-ES, this will not scale to larger uncertainty set dimensions, but provides a safe baseline for optimization performance.

**Worst-case performance.** Table 1 reports the normalized worst-case scores comparing IWOCS, M2TD3, SoftM2TD3, M3DDPG, RARL, and DR using TD3.[6] The worst-case scores for all final

---

[4]Note that these do not respect the rectangularity assumption.

[5]Additional experiments allowing more samples to SAC at each iteration of IWOCS showed only marginal performance gains. This also illustrates how IWOCS can accomodate sub-optimal value functions and policies.

[6]Note that this DR agent is independent of the one we use as a default policy for IWOCS.

Table 1: Avg. of normalized worst-case performance over 10 seeds for each method (HC=half-cheetah, H=hopper, HS=humanoid-standup, IP=inverted-pendulum, W=walker).

| Env | M2TD3 | SoftM2TD3 | M3DDPG | RARL | DR (TD3) | IWOCS* | IWOCS |
|---|---|---|---|---|---|---|---|
| Ant 2 | $\mathbf{1.00 \pm 0.04}$ | $0.92 \pm 0.06$ | $-0.72 \pm 0.05$ | $-1.32 \pm 0.04$ | $0.02 \pm 0.05$ | $0.27 \pm 0.36$ | $-0.27 \pm 0.44$ |
| Ant 3 | $\mathbf{1.00 \pm 0.09}$ | $0.97 \pm 0.18$ | $-0.36 \pm 0.20$ | $-1.28 \pm 0.06$ | $0.61 \pm 0.03$ | $0.28 \pm 0.24$ | $0.43 \pm 0.68$ |
| HC2 | $1.00 \pm 0.05$ | $1.07 \pm 0.05$ | $-0.02 \pm 0.02$ | $-0.05 \pm 0.02$ | $0.84 \pm 0.04$ | $\mathbf{1.13 \pm 0.02}$ | $0.75 \pm 0.28$ |
| HC3 | $1.00 \pm 0.14$ | $\mathbf{1.39 \pm 0.15}$ | $-0.03 \pm 0.05$ | $-0.13 \pm 0.05$ | $1.10 \pm 0.04$ | $0.61 \pm 0.05$ | $0.56 \pm 0.15$ |
| H2 | $1.00 \pm 0.05$ | $1.09 \pm 0.06$ | $0.46 \pm 0.06$ | $0.61 \pm 0.17$ | $0.87 \pm 0.03$ | $\mathbf{6.52 \pm 0.01}$ | $6.34 \pm 0.11$ |
| H3 | $1.00 \pm 0.09$ | $0.68 \pm 0.08$ | $0.22 \pm 0.04$ | $0.56 \pm 0.17$ | $0.73 \pm 0.13$ | $\mathbf{4.94 \pm 0.17}$ | $4.64 \pm 0.16$ |
| HS2 | $1.00 \pm 0.12$ | $\mathbf{1.25 \pm 0.16}$ | $0.98 \pm 0.12$ | $0.88 \pm 0.13$ | $1.14 \pm 0.14$ | $1.02 \pm 0.12$ | $0.98 \pm 0.25$ |
| HS3 | $1.00 \pm 0.11$ | $0.96 \pm 0.07$ | $0.97 \pm 0.07$ | $0.88 \pm 0.13$ | $0.86 \pm 0.06$ | $\mathbf{1.18 \pm 0.08}$ | $1.12 \pm 0.21$ |
| IP2 | $1.00 \pm 0.37$ | $0.38 \pm 0.08$ | $-0.00 \pm 0.00$ | $-0.00 \pm 0.00$ | $0.15 \pm 0.01$ | $\mathbf{2.82 \pm 0.00}$ | $\mathbf{2.82 \pm 0.00}$ |
| W2 | $1.00 \pm 0.14$ | $0.83 \pm 0.15$ | $0.04 \pm 0.04$ | $-0.08 \pm 0.01$ | $0.71 \pm 0.17$ | $\mathbf{1.34 \pm 0.02}$ | $1.23 \pm 0.10$ |
| W3 | $1.00 \pm 0.23$ | $1.03 \pm 0.20$ | $0.06 \pm 0.05$ | $-0.10 \pm 0.01$ | $0.65 \pm 0.19$ | $\mathbf{2.33 \pm 0.10}$ | $2.10 \pm 0.50$ |
| Agg. | $1.0 \pm 0.13$ | $0.96 \pm 0.11$ | $0.15 \pm 0.06$ | $-0.0 \pm 0.07$ | $0.7 \pm 0.08$ | $\mathbf{2.04 \pm 0.11}$ | $1.88 \pm 0.26$ |

policies are evaluated by defining a uniform grid over the transition function's parameter space and performing roll-outs for each transition model. To obtain comparable metrics across environments, we normalize each method's score $v$ using the vanilla TD3 (trained on the default transition function only) reference score $v_{TD3}$ as a minimal baseline and the M2TD3 score $v_{M2TD3}$ as target score: $(v - v_{TD3})/|v_{M2TD3} - v_{TD3}|$. Hence this metric reports how much a method improves upon TD3, compared to how much M2TD3 improved upon TD3. Non-normalized scores are reported in Appendix K. IWOCS* and IWOCS demonstrate competitive performance, outperforming all other methods in 7 out of the 11 environments (note that we did not report results on simpler 1D uncertainty sets). IWOCS* permits a 2.04-fold improvement on average across environments, over the state-of-the-art M2TD3. This validates in practice the soundness of solving a sequence of static models as an alternative to traditional methods building on dynamic models of uncertainty. It seems that Ant is an environment where IWOCS struggles to reach convincing worst case scores. We conjecture this is due to the wide range of possible mass and friction parameters, which make the optimization process very noisy (almost zero mass and friction is a worst-case $T$ making the ant's movement rather chaotic and hence induces a possibly misleading replay buffer) and may prevent the policy optimization algorithm to yield good non-robust policies and value functions given its sample budget. However, IWOCS provides a major (up to 6.5-fold) improvement on other environments.

**Average performance.** While our primary aim is to maximize the worst-case performance, we also appreciate the significance of average performance in real-world scenarios. Table 2 reports the normalized average score (non-normalized scores in Appendix K) obtained by the resulting policy over a uniform grid of 100 transition functions in 2D uncertainty sets (1000 in 3D ones). Interestingly, M3DDPG and RARL feature negative normalized scores and perform worse on average than vanilla TD3 on most environments (as M2TD3 on 3 environments). DR and IWOCS display the highest average performance. Although this outcome was anticipated for DR, it may initially seem surprising for IWOCS, which was not explicitly designed to optimize mean performance. We posit this might be attributed to two factors. First, in MDPs which have not been identified as worst-cases, encountered states are likely to have no valid $Q_{T_j}$ value function. In these MDPs, if we were to apply any of the $\pi_{T_j}$, its score could be as low as the worst cast value (but not lower, otherwise the MDP should have been identified as a worst case earlier). But since IWOCS' indicator functions identify these states as unvisited, the applied policy falls back to the DR policy, possibly providing a slightly better score above the worst case value for these MDPs. Second, the usage of indicator functions permits defining the IWOCS policy as an aggregate of locally optimized policies, possibly avoiding averaging issues. As for the worst-case scores, IWOCS does not perform well on Ant environments. However, it provides significantly better average scores than both DR and M2TD3 on all 9 other benchmarks.

**Worst case paths in the uncertainty set.** IWOCS aims at solving iteratively the robust optimization problem by covering the worst possible case at each iteration. IWOCS and IWOCS* seem to reliably find worst case MDPs and policies in a number of cases, and we could expect the value of the candidate robust policy $\pi_i$ to increase throughout iterations. Table 3 permits tracking the worst case MDPs and policies along iterations for the Humanoid 2D environment (all other results in Appendix M). Specifically, it reports the worst case $T_1 = (\psi_1^0, \psi_1^1)$ for the default policy $\pi_0$, and its

Table 2: Avg. of normalized average performance over 10 seeds for each method.

| Env. | M2TD3 | SoftM2TD3 | M3DDPG | RARL | DR (TD3) | IWOCS* | IWOCS |
|---|---|---|---|---|---|---|---|
| A2 | $1.00 \pm 0.02$ | $1.04 \pm 0.00$ | $-0.13 \pm 0.12$ | $-1.04 \pm 0.02$ | $\mathbf{1.28 \pm 0.03}$ | $1.03 \pm 0.02$ | $1.03 \pm 0.02$ |
| A3 | $-1.00 \pm 0.44$ | $-0.36 \pm 0.46$ | $-6.98 \pm 0.44$ | $-8.94 \pm 0.18$ | $\mathbf{0.92 \pm 0.22}$ | $-0.24 \pm 0.58$ | $-1.06 \pm 2.00$ |
| HC2 | $1.00 \pm 0.03$ | $1.10 \pm 0.04$ | $-1.08 \pm 0.07$ | $-1.94 \pm 0.03$ | $1.84 \pm 0.09$ | $\mathbf{2.12 \pm 0.02}$ | $2.11 \pm 0.04$ |
| HC3 | $1.00 \pm 0.07$ | $1.17 \pm 0.03$ | $-1.43 \pm 0.14$ | $-2.48 \pm 0.05$ | $2.33 \pm 0.12$ | $\mathbf{2.86 \pm 0.02}$ | $2.82 \pm 0.12$ |
| H2 | $1.00 \pm 0.07$ | $0.74 \pm 0.12$ | $-0.40 \pm 0.11$ | $1.86 \pm 0.92$ | $0.36 \pm 0.08$ | $\mathbf{2.27 \pm 0.00}$ | $2.26 \pm 0.07$ |
| H3 | $-1.00 \pm 0.23$ | $-1.20 \pm 0.13$ | $-2.20 \pm 0.37$ | $1.63 \pm 1.53$ | $1.17 \pm 0.23$ | $\mathbf{7.33 \pm 0.30}$ | $7.30 \pm 0.23$ |
| HS2 | $-1.00 \pm 0.67$ | $0.67 \pm 0.83$ | $-1.83 \pm 0.67$ | $-3.00 \pm 1.33$ | $0.50 \pm 0.67$ | $\mathbf{5.33 \pm 1.83}$ | $5.00 \pm 4.00$ |
| HS3 | $1.00 \pm 0.75$ | $0.38 \pm 0.37$ | $0.00 \pm 0.50$ | $-1.75 \pm 0.87$ | $0.38 \pm 0.87$ | $\mathbf{7.75 \pm 0.25}$ | $\mathbf{7.75 \pm 0.25}$ |
| IP2 | $1.00 \pm 0.68$ | $1.06 \pm 0.46$ | $-1.10 \pm 0.25$ | $-0.47 \pm 0.32$ | $2.47 \pm 0.03$ | $\mathbf{2.86 \pm 0.00}$ | $\mathbf{2.86 \pm 0.00}$ |
| W2 | $1.00 \pm 0.06$ | $0.83 \pm 0.16$ | $-0.53 \pm 0.11$ | $-1.21 \pm 0.02$ | $0.91 \pm 0.15$ | $\mathbf{1.13 \pm 0.00}$ | $\mathbf{1.14 \pm 0.01}$ |
| W3 | $1.00 \pm 0.13$ | $0.96 \pm 0.18$ | $-0.57 \pm 0.09$ | $-1.43 \pm 0.04$ | $1.13 \pm 0.10$ | $1.94 \pm 0.05$ | $\mathbf{1.95 \pm 0.05}$ |
| Agg. | $0.45 \pm 0.29$ | $0.58 \pm 0.25$ | $-1.48 \pm 0.26$ | $-1.71 \pm 0.48$ | $1.21 \pm 0.24$ | $\mathbf{3.13 \pm 0.28}$ | $3.01 \pm 0.62$ |

Table 3: Humanoid standup 2, worst parameters search for each iteration over 10 seeds.

| | $\psi_1^0$ | $\psi_1^1$ | $J_{T_1}^{\pi_0}$ | $J_{T_1}^{\pi_1}$ | $\psi_2^0$ | $\psi_2^1$ | $J_{T_2}^{\pi_1}$ | $J_{T_2}^{\pi_2}$ | $\psi_3^0$ | $\psi_3^1$ | $J_{T_3}^{\pi_2}$ | $J_{T_3}^{\pi_3}$ | $\psi_4^0$ | $\psi_4^1$ | $J_{T_4}^{\pi_3}$ |
|---|---|---|---|---|---|---|---|---|---|---|---|---|---|---|---|
| 0 | 7.21 | 14.41 | 5.67 | 12.25 | 0.1 | 11.23 | 7.12 | 11.97 | 0.1 | 9.64 | 7.46 | 13.24 | 0.1 | 9.64 | 7.46 |
| 1 | 7.21 | 14.41 | 5.67 | 6.99 | 0.1 | 8.05 | 7.32 | 12.66 | 0.1 | 8.05 | 7.32 | - | - | - | - |
| 2 | 7.21 | 14.41 | 5.67 | 6.99 | 0.1 | 9.64 | 7.46 | 2.9 | 0.1 | 9.64 | 7.46 | - | - | - | - |
| 3 | 7.21 | 14.41 | 5.67 | 14.47 | 4.84 | 14.41 | 6.54 | 7.39 | 4.84 | 14.41 | 7.46 | - | - | - | - |
| 4 | 7.21 | 14.41 | 5.67 | 14.66 | 0.1 | 12.82 | 7.03 | 8.31 | 0.1 | 12.82 | 7.03 | - | - | - | - |
| 5 | 7.21 | 14.41 | 5.67 | 10.06 | 6.42 | 14.41 | 6.88 | 9.31 | 6.42 | 14.41 | 6.88 | - | - | - | - |
| 6 | 7.21 | 14.41 | 5.67 | 15.81 | 7.21 | 14.41 | 5.67 | - | - | - | - | - | - | - | - |
| 7 | 7.21 | 14.41 | 5.67 | 15.35 | 7.21 | 14.41 | 5.67 | - | - | - | - | - | - | - | - |
| 8 | 7.21 | 14.41 | 5.67 | 13.80 | 7.21 | 14.41 | 5.67 | - | - | - | - | - | - | - | - |
| 9 | 7.21 | 14.41 | 5.67 | 15.16 | 7.21 | 14.41 | 5.67 | - | - | - | - | - | - | - | - |

score $J_{T_1}^{\pi_0}$ (all scores are divided by $10^4$ for readability). The next set of columns repeats these values for later iterations. Each line corresponds to a different random seed. In all lines, the worst case value $J_{T_{i+1}}^{\pi_i}$ steadily increases until convergence. In some runs (last 4 seeds), IWOCS's stopping criterion is met after a single iteration: the worst case $T_2 = (7.21, 14.41)$ for $\pi_1$ is the same as $T_1$, upon which $\pi_1$ was trained. Overall the path through transition models and candidate robust policies illustrates the algorithmic behavior of IWOCS. As indicated in Section 4, IWOCS is guaranteed to converge but may miss the optimal $(T, \pi)$ pair because the selection criterion for $T_{i+1}$ is a (well motivated) heuristic. This happens for a few seeds, where IWOCS seems to reach a different saddle point after 2 or 3 iterations.

## 6 CONCLUSION

The search for robust policies in uncertain MDPs is a long-standing challenge. In this work, we proposed to revisit the static model of transition function uncertainty, which is equivalent to the dynamic model in the case of $sa$-rectangular uncertainty sets and stationary policies. We proposed to exploit this equivalence and the no-duality-gap property to design an algorithm that trades the resolution of a two-player game, for a sequence of one-player MDP resolutions. This led to the IWOCS (Incremental Worst-Case Search) meta-algorithm, which incrementally builds a discrete, non-$sa$-rectangular uncertainty set and a sequence of candidate robust policies. An instance of IWOCS, using SAC for policy optimization, and CMA-ES or grid-search for worst-case search, appeared as a relevant method on popular robust RL benchmarks, and outperformed the state-of-the-art algorithms on a number of environments. IWOCS proposes a new perspective on the resolution of robust MDPs and robust RL problems, which appears as a competitive formulation with respect to traditional methods. It also poses new questions, like the tradeoffs between policy optimization precision and overall robustness, gradient-based methods for worst-case search, bounds due to approximate value functions, or validity of using $Q_i$ as a surrogate of the robust value function for the $\mathcal{T}_i$ uncertainty set. All these open interesting avenues for future research.

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

# A   APPENDIX

# B   KEY RESULTS FROM THE LITERATURE

In order to ease the reading of this paper, we recall the two theoretical results that Section 2 builds upon. We reproduce the text from the original papers (Iyengar, 2022; Wiesemann et al., 2013) but, for the sake of consistency, we use the notations of the present paper and indicate [in brackets] whenever we adjusted the original notation. All results quoted below apply to $sa$-rectangular uncertainty sets.

**Definition of the static and dynamic models, in the introduction of section 3 of (Iyengar, 2005).**

> (i) Static model: The adversary is restricted to choose the same, but unknown, $[T(\cdot|s,a)]$ every time the state-action pair $(s,a)$ is encountered.
> (ii) Dynamic model: The adversary is allowed to choose a possibly different conditional measure $[T(\cdot|s,a)]$ every time the state-action pair $(s,a)$ is encountered.
> [...]
> As mentioned in the introduction, the goal of the robust formulation is to systematically mitigate the effect of errors associated with estimating the state transitions; i.e., the state transition is, in fact, fixed but the decision maker is only able to estimate it to within a set. Thus, the static model is appropriate for this scenario. However, computing the optimal policy for the static model is NP-hard, therefore, we will restrict attention to the dynamic model. Clearly the value function in the dynamic model is a lower bound for the value function in the static model. We contrast the implications of the two models in Lemma 3.3.

**Equivalence of the value functions under the static and dynamic models (Iyengar, 2005, Lemma 3.3).**

> Lemma 3.3 (Dynamic vs. static adversary). Let $[\pi : \mathcal{S} \to \mathcal{A}$ be a] stationary policy. Let $V^\pi$ and $\hat{V}^\pi$ be the value of the $\pi$ in the dynamic and static model respectively.

Then $V^\pi = \hat{V}^\pi$.

[...]

In the proof of the result we have implicitly established that the "best-response" of dynamic adversary when the decision maker employs a stationary policy is, in fact, static [...].

**Non-equivalence of the static and dynamic models for non-stationary policies, at the end of Section 3 of (Iyengar, 2005)**

Lemma 3.3 highlights an interesting asymmetry between the decision maker and the adversary that is a consequence of the fact that the adversary plays second. While it is optimal for a dynamic adversary to play static (stationary) policies when the decision maker is restricted to stationary policies, it is not optimal for the decision maker to play stationary policies against a static adversary.

**No-duality gap property (Wiesemann et al., 2013, Equation 4).**

To date, the literature on robust MDPs has focused on (s, a)-rectangular ambiguity sets. For this class of ambiguity sets, it is shown in (Iyengar, 2005) and (Nilim & El Ghaoui, 2005) that the worst-case expected total reward [...] is maximized by a deterministic stationary policy for finite and infinite horizon MDPs. Optimal policies can be determined via extensions of the value and policy iteration. For some ambiguity sets, finding an optimal policy, as well as evaluating (2) for a given stationary policy, can be achieved in polynomial time. Moreover, the policy improvement problem satisfies the following saddle point condition

$$\sup_\pi \inf_{T \in \mathcal{T}} \mathbb{E}\left[\sum_{t=0}^\infty \gamma^t r(s_t, a_t, s_{t+1})|s_0\right] = \inf_{T \in \mathcal{T}} \sup_\pi \mathbb{E}\left[\sum_{t=0}^\infty \gamma^t r(s_t, a_t, s_{t+1})|s_0\right]$$

## C NOTATIONS

Table 4 recalls all key notations used throughout the paper. The first block in Table 4 is for standard (non-robust) MDP quantities (used in Section 2 and after), the second for standard robust MDP quantities (Section 2 and after), the third for IWOCS-specific quantities (Section 4 and after), and finally the fourth for Deep-IWOCS notations (Section 5 and after).

Table 4: Key notations

| Symbol | Meaning |
|---|---|
| $M_T = (\mathcal{S}, \mathcal{A}, T, r)$ | MDP with transition kernel $T$ |
| $\pi$ | Stationary policy $\mathcal{S} \to \mathcal{A}$ |
| $V_T^\pi, Q_T^\pi$ | State and state-action value functions of policy $\pi$ in $M_T$ |
| $J_T^\pi$ | Scalar value $V_T^\pi(s_0)$ of initial state under $\pi$ in $M_T$ |
| $\pi_T^*$ | Optimal policy in $M_T$ |
| $V_T^*, Q_T^*$ | Optimal state and state-action value functions in $M_T$ |
| $\mathcal{T}$ | Uncertainty set |
| $V_\mathbf{T}^\pi$ | Value function of policy $\pi$ under sequence of transition kernels $\mathbf{T}$ |
| $V_\mathcal{T}^\pi, Q_\mathcal{T}^\pi$ | Pessimistic value function of policy $\pi$ for uncertainty set $\mathcal{T}$ |
| $V_\mathcal{T}^*, Q_\mathcal{T}^*$ | Robust value function for uncertainty set $\mathcal{T}$ |
| $\mathcal{T}_i$ | Non-$sa$-rectangular discrete uncertainty set |
| $\tilde{\mathcal{T}}_i$ | $sa$-rectangular uncertainty superset of $\mathcal{T}_i$ |
| $Q_i$ | $T_i$-robust value function |
| $\pi_T$ | SAC's approximation of $\pi_T^*$ |
| $Q_T$ | SAC's approximation of $Q_T^*$ |
| $Q_T'$ | SAC's estimate of $\pi_T$'s regularized value function |
| $\hat{T}_j$ | Predictive coding model for $T_j$ |

## D  COMPUTING RESOURCES

All experiments were run on a desktop machine (Intel i9, 10th generation processor, 64GB RAM) with a single NVIDIA RTX 3090 GPU. Averages, medians, and standard deviations were computed from 10 independent repetitions of each experiment.

## E  WINDY-WALK GRIDWORLD

The windy-walk environment used in Section 4 is a discrete grid-world environment illustrated in Figure 2. It features 36 discrete states corresponding to positions on the grid, and 4 discrete actions corresponding to cardinal moves. Six states are unreachable and correspond to walls, defining three corridors. The transition model is deterministic by default, except in the corridors where the wind blows. This transition model is parameterized by a scalar parameters $\alpha$. In the Northern corridor:

- the $W$ action moves West with probability 1,
- the $N$ and $S$ actions leave the position unchanged with probability $1 - \alpha$ and the agent is pushed West with probability $\alpha$,
- the $E$ action moves East with probability $1 - \alpha$ and West with probability $\alpha$.

The middle corridor works the same way, but with probability $\alpha^3$ instead of $\alpha$. In the Southern corridor:

- the $W$ action moves West with probability 1,
- the $N$ (resp. $S$) action move the agent respectively North (resp. South) with probability $1 - \alpha^6$ (unless it runs into a wall in which case the position is unchanged), and West with probability $\alpha^6$,
- the $E$ action moves East with probability $1 - \alpha^6$ and West with probability $\alpha^6$.

Rewards are -1 for all transitions and the $G$ state is an absorbing goal states yielding zero reward. The agent always starts in state $S$. Consequently, windy-walk is a stochastic shortest path. For small values of $\alpha$, the optimal policy is to go straight from $S$ to $G$, but as $\alpha$ increases, the wind blows harder, and it becomes more interesting to make a detour through the middle then Southern corridors. The corresponding robust MDP problem features an uncertainty set spanned by 25 discrete values of $\alpha$, uniformly distributed in $[0, 0.5]$.

The instance of IWOCS evaluated in Section 4 uses value iteration as a policy optimization algorithm and a brute-force grid search as a search method for worst-case transition functions, as summarized in Algorithm 1. In Algorithm 1 we abusively write $T_\alpha$ the transition model parameterized by $\alpha$.

In the experiments of Section 4, value iteration is run until a tolerance of $10^{-3}$ is met. $\gamma$ is set to 0.95. Monte-Carlo estimates of $V_T^{\pi_i}(s_0)$ use 300 independent rollouts (of length at most $10^4$) from the starting state.

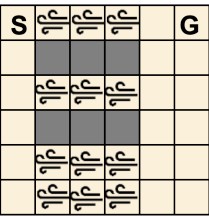

Figure 2: Windy walk grid-world.

## F  SOFT ACTOR-CRITIC AND CMA-ES HYPERPARAMETERS

Deep IWOCS uses SAC (Haarnoja et al., 2018) for policy optimization and trains jointly a predictive coding model to predict the outcome of a state-action pair. Specifically, a single network called

---

**Algorithm 1:** IWOCS with value iteration and brute force worst-case search

---

**Input:** $\mathcal{T}$, $T_0$, max number of iterations $M$, tolerance on robust value $\epsilon$

**for** $i = 0$ *to* $M$ **do**

1    Find $Q^*_{T_i} = \texttt{value\_iteration}(T_i)$    /* Non-robust policy optimization */

2    Define $\mathcal{T}_i = \{T_j\}_{j \in [0,i]}$

3    Define $Q_i : s, a \mapsto \min_{j \in [0,i]}\{Q^*_{T_j}(s,a)\}$    /* $\mathcal{T}_i$-robust value function */

4    Define $\pi_i(s) = \arg\max_a(Q_i(s,a))$    /* Candidate policy */

5    $V^{\pi_i}_{T_{i+1}}(s_0) = +\infty$

6    **for** $\alpha \in \mathcal{T}$ **do**    /* Identify worst $T$ */

7      $\tilde{V} = \texttt{Monte-Carlo\_rollouts}(\pi_i)$

8      **if** $\tilde{V} < V^{\pi_i}_{T_{i+1}}(s_0)$ **then**

9        $V^{\pi_i}_{T_{i+1}}(s_0) = \tilde{V}$

10        $T_{i+1} = T_\alpha$

11    **if** $|V^{\pi_i}_{T_{i+1}}(s_0) - Q_i(s_0, \pi_i(s_0)| \leqslant \epsilon$ **then**

12      **return** $\pi_i$, $T_{i+1}$, $V^{\pi_i}_{T_{i+1}}(s_0)$    /* Early termination condition */

**return** $\pi_M$, $T_{M+1}$, $V^{\pi_M}_{T_{M+1}}(s_0)$

---

"enhanced critic" is trained to predict the regularized value function $Q'(s,a)$, the unregularized value function $Q(s,a)$ and a prediction of the transition outcome $\hat{T}(s,a)$. The critic network's architecture is summarized in Figure 3. All activation functions are ReLU except for the output layers (identity functions). Note that one more layer was necessary to appropriately estimate $Q$ compared to $Q'$. Our implementation also uses double critics as per the common practice, to avoid overestimating $Q$ and $Q'$ (totally independent networks, no shared layers). Given a replay buffer $\mathcal{D}$, learning $Q$ minimizes the loss

$$L_Q = \mathbb{E}_{s,a,s' \sim \mathcal{D}, a' \sim \pi}\left[Q(s,a) - [r + \gamma Q^-(s',a')]\right]^2,$$

where $Q^-$ is a target network, updated through Polyak averaging. Similarly, $Q'$ minimizes

$$L'_Q = \mathbb{E}_{s,a,s' \sim \mathcal{D}, a' \sim \pi}\left[Q'(s,a) - [r + \gamma(Q'^-(s',a') - \alpha \log \pi(a'|s'))]\right]^2.$$

Finally, $\hat{T}$ minimizes

$$L_{\hat{T}} = \mathbb{E}_{s,a,s' \sim \mathcal{D}}\left[\|\hat{T}(s,a) - s'\|_1\right].$$

These three objective functions are minimized in turn with three distinct Adam optimizers to account for possible different orders of magnitude.

The actor network is a standard SAC actor trained with respect to the regularized Q-function $Q'$. The network's architecture is depicted in Figure 4. All activation functions are ReLU, except for the output values (identify for $\mu$ and $\tanh$ for $\log \sigma$ as per the common practice). The output action drawn from the network's output is run through an additional $\tanh$ function following the usual SAC implementations.

The search for worst case transition functions is performed by using the CMA-ES black-box optimization method (Hansen & Ostermeier, 2001). The implementation used is the reference one of https://github.com/CyberAgentAILab/cmaes, off-the-shelf.

All hyperparameter values for SAC and CMA-ES are summarized in Table 5. These values are the same across all experiments.

## G    ADAPTIVE THRESHOLDING FOR PREDICTIVE CODING

As introduced in Section 5, the SAC-based implementation of IWOCS used in the experiments exploits a predictive coding mechanism in order to characterize each policy's training distribution support and to avoid using a given $\pi_{T_i}$ on samples outside its training distribution. Policy $\pi_{T_i}$ and

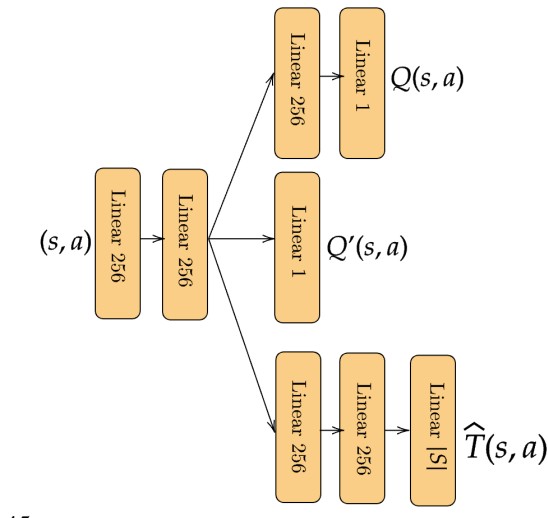

.45

Figure 3: Enhanced critic

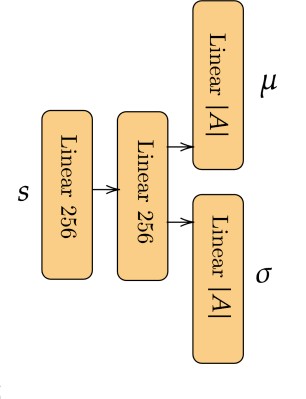

.45

Figure 4: Actor

Figure 5: Network architectures

value function $Q_{T_i}$ are deemed usable in state $s_t$ along the current trajectory if $s_t$ was accurately predicted by the dynamics model $\hat{T}_i(s_{t-1}, a_{t-1})$. Specifically, we consider a threshold $\rho_i$ on the prediction error and consider $Q_{T_i}$ and $\pi_{T_i}$ to be viable in $s_t$ if $\ell(\hat{T}_i(s_{t-1}, a_{t-1}), s_t) \leqslant \rho_i$, with $\ell(\hat{T}_i(s_{t-1}, a_{t-1}), s_t) = \|\hat{T}_i(s_{t-1}, a_{t-1}) - s_t\|_1$. In states where $Q_{T_i}$ is non-viable, we arbitrarily set its value to $+\infty$ so that it does not participate in $Q_i(s, a) = \min_{j \leqslant i}\{Q_{T_j}(s, a)\}$. We noted in the main text that alternative characterizations of the support distribution were possible, and we do not claim the present choice outperforms the alternatives. Notably, all choices induce a number of parameters to tune (here $\rho_i$). This leads to a number of design choices that make the implementation somehow more convoluted than the simple principle of IWOCS. While the main text kept things focused on the principles of IWOCS, we provide here a full pseudo-code (which is more representative of the provided code) for the sake of completeness. Appendix F already covered the network structure, the training losses and the training hyperparameters for SAC and CMA-ES. Hence, the present section focuses on how to adjust each $\rho_i$.

Training of the enhanced critic network does not provide a usable value for $\rho_i$ and experimental results demonstrated that accurate characterization of the training distribution's support required per-MDP tuning. Since $\rho_i$ needs to be tuned for $\pi_{T_i}$ during iteration $i$ (and is kept fixed thereafter), we couple its search with that of the worst case transition model to permit better overall efficiency.

Table 5: Hyperparameters of SAC

| Hyperparameter | Value |
|---|---|
| Learning rate actor | 3e-4 |
| Learning rate critic | 1e-3 |
| Adam epsilon | 1e-5 |
| Adam $(\beta_1, \beta_2)$ | (0.9, 0.999) |
| Batch size | 256 |
| Memory size | 1e6 |
| Gamma | 0.99 |
| Polyak update | 0.995 |
| Number of steps before training | 1e4 |
| CMA-ES generations | 6 |
| CMAS-ES population size | 100 |
| CMA ES mean | 0.5 |
| CMA ES std | 0.5 |

Specifically, at iteration $i$, we consider a discrete set $R$ of possible values for threshold $\rho_i$. For each value in $R$, we identify the worst case transition model. Then, we keep the value of $\rho_i$ that enabled the best pessimistic value. In a sense, this makes $\rho_i$ a parameter of the candidate robust policy. This parameter is single-dimensional and hence its optimization is computationally undemanding. We emphasize that this tuning mechanism is both very naive and arbitrary. It is naive since it performs a grid search over discrete values of $\rho_i$, where it could have exploited optimization methods. It is arbitrary in the sense that it picks $\rho_i$ by keeping as a selection heuristic the overall goal of identifying robust policies.

Algorithm 2 summarizes the complete IWOCS process with adaptive thresholding for predictive coding. In the experiments of Section 5, $R$ is a discrete set of 10 values evenly spaced between 0.1 and 1.

## H  SAMPLE BUDGETS

In order to enable a fair comparison with the results of Tanabe et al. (2022) which we report in Table 1, we evaluate IWOCS with the same overall sample budget, ie. 4 million samples in 2D uncertainty set environments and 5 million samples in 3D uncertainty set environments.

In 2D environments, the default DR policy is trained for $1.6 \cdot 10^6$ steps, then 3 IWOCS iterations of $8 \cdot 10^5$ each are run, for a total of $4 \cdot 10^6$ collected samples.

In 3D environments, the default DR policy is trained for $1.8 \cdot 10^6$ steps, then 4 IWOCS iterations of $8 \cdot 10^5$ each are run, for a total of $5 \cdot 10^6$ collected samples.

No fine-tuning of these training durations was performed.

## I  COMPUTATIONAL OVERHEAD DUE TO IWOCS

In Table 6 we report the average wall-clock time needed for our implementation of SAC to cover the 4 (resp. 5) million samples allocated for 2D (resp. 3D) environments without IWOCS. Then, we report the time required by IWOCS to cover the same sample budget. This permits a fair evaluation of the overhead computational cost of IWOCS, without the bias due to implementation optimizations.

## J  UNCERTAINTY SETS IN MUJOCO ENVIRONMENTS

The experiments of Section 5 follow the evaluation protocol proposed by Tanabe et al. (2022) and based on MuJoCo environments (Todorov et al., 2012). These environments are designed with 2D or 3D uncertainty sets. Table 7 lists all environments evaluated, along with their uncertainty sets. The

---

**Algorithm 2:** Deep IWOCS with adaptive threshold

**Input:** $\mathcal{T}$, $T_0$, maximum number of iterations $M$, discrete thresholds set $R$

**for** $i = 0$ *to* $M$ **do**

  Find $Q_{T_i}, \pi_{T_i}, \hat{T}_i = \text{SAC}(T_i)$      // Non-robust SAC and pred coding

1   Define $\mathcal{T}_i = \{T_j\}_{j\in[0,i]}$

2   Define $\hat{V} = -\infty$      // candidate worst value

3   **for** $\rho \in R$ **do**      // loop over thresholds

4    Define $\tilde{Q}_{T_i}(s,a) = \begin{cases} +\infty \text{ if } \ell(\hat{T}_i(s_{t-1}, a_{t-1}); s_t) > \rho \text{ for } s = s_t \\ Q_{T_i}(s,a) \text{ otherwise.} \end{cases}$

5    Define $\tilde{Q}_i(s,a) = \min_{j\leqslant i}\{\tilde{Q}_{T_j}(s,a)\}_{j\in[0,i]}, \forall s, a$      // $\mathcal{T}_i$-robust value function

6    Define $J^*(s) = \arg\min_{j\leqslant i} \tilde{Q}_{T_j}(s, \pi_{T_j}(s)), \forall s$

7    Define $\tilde{\pi}_i(s) = \begin{cases} \pi_{default}(s) \text{ if } \tilde{Q}_i(s,a) = +\infty, \\ \pi_{T_{j^*}}(s) \text{ with } j^* \in J^*(s) \text{ otherwise.} \end{cases}$      // Candidate policy

8    Find $\tilde{T}_{i+1}, V^{\tilde{\pi}_i}_{\tilde{T}_{i+1}}(s_0) = \text{CMA-ES}(V^{\tilde{\pi}_i}_T(s_0))$      // Identify worst $T$

9    **if** $V^{\tilde{\pi}_i}_{\tilde{T}_{i+1}}(s_0) \geqslant \hat{V}$ **then**      // keep best $\rho$

10     Set $\rho_j = \rho, \hat{V} = V^{\tilde{\pi}_i}_{\tilde{T}_{i+1}}(s_0)$

11     Define $T_{i+1} = \tilde{T}_{i+1}, \pi_i = \tilde{\pi}_i, V^{\pi_i}_{T_{i+1}}(s_0) = V^{\tilde{\pi}_i}_{\tilde{T}_{i+1}}(s_0), Q_i = \tilde{Q}_i$

12   **if** $|V^{\pi_i}_{T_{i+1}}(s_0) - Q_i(s_0, \pi_i(s_0)| \leqslant \epsilon$ **then**

13    **return** $\pi_i, T_{i+1}, V^{\pi_i}_{T_{i+1}}(s_0)$      // Early termination condition

**return** $\pi_M, T_{M+1}, V^{\pi_M}_{T_{M+1}}(s_0)$

---

| Environment | SAC | IWOCS |
|---|---|---|
| Ant 2 | 18h | 36h |
| Ant 3 | 22.5h | 40h |
| Halfcheetah 2 | 20h | 38.5h |
| Halfcheetah 3 | 25h | 41h |
| Walker 2 | 19h | 40h |
| Walker 3 | 24h | 45h |
| Hopper 2 | 20h | 38h |
| Hopper 3 | 25h | 47h |
| HumanoidStandup 2 | 18h | 40h |
| HumanoidStandup 3 | 22.5h | 48h |

Table 6: Average wall-clock time for plain SAC and for IWOCS for the same number of samples.

uncertainty sets column defines the ranges of variation for the parameters within each environment. The reference parameters column indicates the nominal or default values. The uncertainty parameters column describes the physical meaning of each parameter.

Table 7: List of environment and parameters for the experiements

| Environment | Uncertainty set $\mathcal{T}$ | Reference values | Uncertainty parameters |
|---|---|---|---|
| Ant 2 | $[0.1, 3.0] \times [0.01, 3.0]$ | $(0.33, 0.04)$ | torso mass; front left leg mass |
| Ant 3 | $[0.1, 3.0] \times [0.01, 3.0] \times [0.01, 3.0]$ | $(0.33, 0.04, 0.06)$ | torso mass; front left leg mass; front right leg mass |
| HalfCheetah 2 | $[0.1, 4.0] \times [0.1, 7.0]$ | $(0.4, 6.36)$ | world friction; torso mass |
| HalfCheetah 3 | $[0.1, 4.0] \times [0.1, 7.0] \times [0.1, 3.0]$ | $(0.4, 6.36, 1.53)$ | world friction; torso mass; back thigh mass |
| Hopper 2 | $[0.1, 3.0] \times [0.1, 3.0]$ | $(1.00, 3.53)$ | world friction; torso mass |
| Hopper 3 | $[0.1, 3.0] \times [0.1, 3.0] \times [0.1, 4.0]$ | $(1.00, 3.53, 3.93)$ | world friction; torso mass; thigh mass |
| HumanoidStandup 2 | $[0.1, 16.0] \times [0.1, 8.0]$ | $(8.32, 1.77)$ | torso mass; right foot mass |
| HumanoidStandup 3 | $[0.1, 16.0] \times [0.1, 5.0] \times [0.1, 8.0]$ | $(8.32, 1.77, 4.53)$ | torso mass; right foot mass; left thigh mass |
| InvertedPendulum 2 | $[1.0, 31.0] \times [1.0, 11.0]$ | $(4.90, 9.42)$ | pole mass; cart mass |
| Walker 2 | $[0.1, 4.0] \times [0.1, 5.0]$ | $(0.7, 3.53)$ | world friction; torso mass |
| Walker 3 | $[0.1, 4.0] \times [0.1, 5.0] \times [0.1, 6.0]$ | $(0.7, 3.53, 3.93)$ | world friction; torso mass; thigh mass |

Table 8: Avg. $\pm$ std. error of worst-case performance over 10 seeds for each method

| Environment | M2TD3 | SoftM2TD3 | M3DDPG | RARL | DR (TD3) | IWOCS* | IWOCS |
|---|---|---|---|---|---|---|---|
| Ant 2 | 4.13 ± 0.11 | 3.92 ± 0.14 | -0.25 ± 0.13 | -1.77 ± 0.09 | 1.64 ± 0.13 | 2.27 ± 0.91 | 0.90 ± 1.13 |
| Ant 3 | 0.10 ± 0.10 | 0.07 ± 0.20 | -1.38 ± 0.22 | -2.38 ± 0.07 | -0.32 ± 0.03 | -0.69 ± 0.26 | -0.52 ± 0.74 |
| HalfCheetah 2 | 2.61 ± 0.16 | 2.82 ± 0.16 | -0.58 ± 0.06 | -0.70 ± 0.05 | 2.12 ± 0.13 | 3.02 ± 0.07 | 1.81 ± 0.87 |
| HalfCheetah 3 | 0.93 ± 0.21 | 1.53 ± 0.23 | -0.66 ± 0.08 | -0.81 ± 0.07 | 1.09 ± 0.06 | 0.33 ± 0.07 | 0.25 ± 0.23 |
| Hopper 2 | 5.33 ± 0.28 | 5.79 ± 0.29 | 2.58 ± 0.29 | 3.34 ± 0.89 | 4.68 ± 0.15 | 33.58 ± 0.03 | 32.68 ± 0.54 |
| Hopper 3 | 2.84 ± 0.25 | 1.98 ± 0.22 | 0.73 ± 0.11 | 1.64 ± 0.46 | 2.10 ± 0.35 | 13.47 ± 0.45 | 12.66 ± 0.42 |
| HumanoidStandup 2 | 6.50 ± 0.70 | 7.94 ± 0.90 | 6.37 ± 0.72 | 5.78 ± 0.73 | 7.31 ± 0.78 | 6.62 ± 0.71 | 6.41 ± 1.46 |
| HumanoidStandup 3 | 6.20 ± 0.64 | 5.99 ± 0.37 | 6.01 ± 0.38 | 5.54 ± 0.76 | 5.41 ± 0.34 | 7.19 ± 0.46 | 6.86 ± 1.19 |
| InvertedPendulum 2 | 3.56 ± 1.32 | 1.36 ± 0.30 | 0.02 ± 0.00 | 0.02 ± 0.00 | 0.57 ± 0.02 | 10 ± 00 | 10 ± 00 |
| Walker 2 | 3.14 ± 0.39 | 2.64 ± 0.43 | 0.39 ± 0.11 | 0.06 ± 0.04 | 2.31 ± 0.50 | 4.10 ± 0.07 | 3.80 ± 0.28 |
| Walker 3 | 1.94 ± 0.40 | 2.00 ± 0.35 | 0.28 ± 0.09 | 0.00 ± 0.02 | 1.32 ± 0.34 | 4.29 ± 0.18 | 3.89 ± 0.89 |

Table 9: Avg. $\pm$ std. deviation of average performance over 10 seeds for each method

| Environment | M2TD3 | SoftM2TD3 | M3DDPG | RARL (DDPG) | DR (TD3) | IWOCS* | IWOCS |
|---|---|---|---|---|---|---|---|
| Ant 2 | 5.44 ± 0.05 | 5.56 ± 0.01 | 1.86 ± 0.38 | -1.00 ± 0.06 | 6.32 ± 0.09 | 5.54 ± 0.06 | 5.53 ± 0.06 |
| Ant 3 | 2.66 ± 0.22 | 2.98 ± 0.23 | -0.33 ± 0.22 | -1.31 ± 0.09 | 3.62 ± 0.11 | 3.04 ± 0.29 | 2.63 ± 1.00 |
| HalfCheetah 2 | 4.35 ± 0.05 | 4.52 ± 0.07 | 0.77 ± 0.12 | -0.70 ± 0.05 | 5.79 ± 0.15 | 6.28 ± 0.04 | 6.26 ± 0.07 |
| HalfCheetah 3 | 3.79 ± 0.09 | 4.02 ± 0.04 | 0.58 ± 0.18 | -0.81 ± 0.07 | 5.54 ± 0.16 | 6.24 ± 0.03 | 6.19 ± 0.16 |
| Hopper 2 | 2.51 ± 0.07 | 2.26 ± 0.12 | 1.15 ± 0.11 | 3.34 ± 0.89 | 1.89 ± 0.08 | 3.74 ± 0.00 | 3.73 ± 0.07 |
| Hopper 3 | 0.85 ± 0.07 | 0.79 ± 0.04 | 0.49 ± 0.11 | 1.64 ± 0.46 | 1.50 ± 0.07 | 3.35 ± 0.09 | 3.34 ± 0.07 |
| HumanoidStandup 2 | 0.97 ± 0.04 | 1.07 ± 0.05 | 0.92 ± 0.04 | 0.85 ± 0.08 | 1.06 ± 0.04 | 1.35 ± 0.11 | 1.33 ± 0.24 |
| HumanoidStandup 3 | 1.09 ± 0.06 | 1.04 ± 0.03 | 1.01 ± 0.04 | 0.87 ± 0.07 | 1.04 ± 0.07 | 1.63 ± 0.02 | 1.63 ± 0.02 |
| InvertedPendulum 2 | 6.13 ± 1.42 | 6.26 ± 0.95 | 1.76 ± 0.51 | 3.07 ± 0.66 | 9.18 ± 0.07 | 10 ± 00 | 10 ± 00 |
| Walker 2 | 4.72 ± 0.12 | 4.37 ± 0.32 | 1.63 ± 0.22 | 0.26 ± 0.05 | 4.54 ± 0.31 | 4.99 ± 0.01 | 5.00 ± 0.02 |
| Walker 3 | 4.27 ± 0.21 | 4.21 ± 0.30 | 1.65 ± 0.15 | 0.21 ± 0.07 | 4.48 ± 0.16 | 5.84 ± 0.08 | 5.86 ± 0.08 |

## K    Non-normalized results

Table 8 reports the non-normalized worst case scores, averaged across 10 independent runs for each benchmark. Table 9 reports the average score obtained by each agent across a grid of environments, also averaged across 10 independent runs for each benchmark.

## L    Soft Actor Critic baseline

We conducted additional experiments using SAC to train on the reference transition kernel across all environments. These experiments aimed to confirm the performance similarities with TD3 and to emphasize the performance gap with IWOCS. The results are summarized in Table L.

Overall, SAC performs poorly in terms of worst-case score, reinforcing our earlier observation of its similarity to TD3. Specifically, SAC exhibits variability across different environments, sometimes outperforming TD3 and sometimes not. However, the scores generally remain within the same order of magnitude. For instance, in the Hopper environment, SAC achieves scores of 14 and 5.8 on Hopper 2 and Hopper 3, respectively, which surpass previous state-of-the-art methods. Despite this, IWOCS still shows a significant improvement, with final scores of 32.68 and 12.66, respectively. These results highlight the distinct advantages of our approach in achieving robust performance.

## M    Worst-case paths

Table 3 illustrated the path followed by the successive identified worst-case transition functions $T_i$ in the 2D uncertainty set of the Humanoid Standup 2 environment, across 10 independent optimization runs. For the sake of completeness, we provide here the same results for all environments, which permit drawing similar conclusions. Tables 11 and 12 start by recalling the physical meaning of each transition function's parameters. Then, Tables 13 to 21 follow the same logic as Table 3 and report the evolution of worst-case parameters and values on all other environments than Humanoid Standup 2.

| Environment | Avg ± Std |
|---|---|
| Ant 2 | -1.0 ± 0.4 |
| Ant 3 | -2.1 ± 0.1 |
| Halfcheetah 2 | -0.2 ± 0.002 |
| Halfcheetah 3 | -0.26 ± 0.04 |
| Hopper 2 | 14 ± 0.99 |
| Hopper 3 | 5.8 ± 0.4 |
| Humanoid 2 | 3.8 ± 0.1 |
| Humanoid 3 | 3.6 ± 0.27 |
| Walker 2 | 0.8 ± 0.9 |
| Walker 3 | 0.5 ± 0.7 |

Table 10: Avg $\pm$ std. error of worst-case performance over 10 for SAC

Table 11: Physical meaning of transition function parameters in 2D environments

| Environment | $\psi_1$ | $\psi_2$ |
|---|---|---|
| Ant 2 | torso mass | front left leg mass |
| Halfcheetah 2 | world friction | torso mass |
| Hopper 2 | world friction | torso mass |
| HumanoidStandup 2 | right foot mass | torsomass |
| Walker 2 | world friction | torso mass |

## N  HOW MANY VALID POLICIES IN $s$?

The Deep-IWOCS method proposed in Section 5 introduced indicator functions constraining the use of a given policy to a subset of states. Depending on the environment and uncertainty parameters, we expect some policies to remain within the same set of explored states, while others will cover a very different state distribution. To quantify this aspect, we ran an experiment where the IWOCS final policy is run on the Hopper 3 benchmark, across a grid of transition functions. For each encountered state, we count how many policies are valid. Figure 6 reports the corresponding histograms (note the log-scale on the $y$-axis).

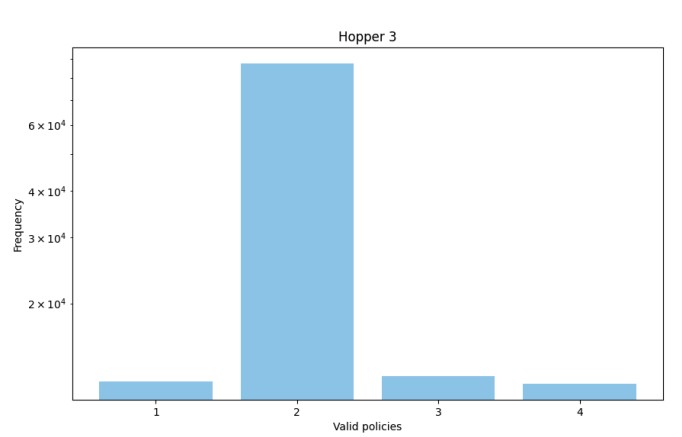

Figure 6: Counting how many policies are valid in each state, in Hopper 3

## O  ON THE VARIANCE OF THE DOMAIN RANDOMIZATION POLICY

Domain randomization on the full uncertainty set yields policies with a very large span of worst-case scores from one random seed to the other (Table 22). In other words, DR provides policies with a very large variance in worst-case performance. In turn, running a separate DR training for each

Table 12: Physical meaning of transition function parameters in 3D environments

| Environment | $\psi_1$ | $\psi_2$ | $\psi_3$ |
|---|---|---|---|
| Ant 3 | torso mass | front left leg mass | front right leg mass |
| Halfcheetah 3 | world friction | torso mass | back thigh mass |
| Hopper 3 | world friction | torso mass | thigh mass |
| HumanoidStandup 3 | torso mass | left thigh mass | right foot mass |
| Walker 3 | world friction | torso mass | thigh mass |

Table 13: Ant 2, worst parameters search for each iteration over 10 seeds.

| | $\psi_1^0$ | $\psi_1^1$ | $J_{T_1}^{\pi_0}$ | $J_{T_1}^{\pi_1}$ | $\psi_2^0$ | $\psi_2^1$ | $J_{T_2}^{\pi_1}$ | $J_{T_2}^{\pi_2}$ | $\psi_3^0$ | $\psi_3^1$ | $J_{T_3}^{\pi_2}$ | $J_{T_3}^{\pi_3}$ | $\psi_4^0$ | $\psi_4^1$ | $J_{T_4}^{\pi_3}$ |
|---|---|---|---|---|---|---|---|---|---|---|---|---|---|---|---|
| 0 | 0.68 | 0.608 | 0.5 | 2.87 | 0.1 | 0.01 | 2.5 | 2.44 | 0.1 | 0.01 | 2.5 | - | - | - | - |
| 1 | 0.68 | 0.608 | 0.5 | 3.58 | 0.1 | 0.309 | 2.5 | 2.8 | 0.1 | 0.01 | 2.5 | - | - | - | - |
| 2 | 0.68 | 0.608 | 0.5 | 0.06 | 0.1 | 0.01 | 2.5 | 2.32 | 0.1 | 0.01 | 2.5 | - | - | - | - |
| 3 | 0.68 | 0.608 | 0.5 | 3.54 | 0.1 | 0.01 | 2.67 | 2.17 | 0.1 | 0.01 | 2.67 | - | - | - | - |
| 4 | 0.68 | 0.608 | 0.5 | 5.13 | 0.1 | 0.01 | 2.47 | 0.69 | 0.1 | 0.01 | 2.47 | - | - | - | - |
| 5 | 0.68 | 0.608 | 0.5 | 1.77 | 0.1 | 0.01 | 3.56 | 0.51 | 0.1 | 0.01 | 3.56 | - | - | - | - |
| 6 | 0.68 | 0.608 | 0.5 | 4.81 | 0.1 | 2.103 | 0.83 | 2.1 | 0.1 | 2.103 | 0.83 | - | - | - | - |
| 7 | 0.68 | 0.608 | 0.5 | 0.4 | 0.1 | 2.402 | 1.81 | 1.91 | 0.1 | 0.01 | 2.55 | 1.3 | 0.39 | 0.01 | 2.55 |
| 8 | 0.68 | 0.608 | 0.5 | 0.03 | 0.1 | 0.01 | 2.5 | 3.1 | 0.1 | 0.01 | 2.5 | - | - | - | - |
| 9 | 0.68 | 0.608 | 0.5 | 4.88 | 0.1 | 0.01 | 2.51 | 2.3 | 0.1 | 0.01 | 2.51 | - | - | - | - |

seed of IWOCS induces a large variance on scores across seeds from the first iteration. IWOCS still converges, but this variance is carried through the iterations. For the sake of completeness, Table 23 report the scores of IWOCS and IWOCS* with a varying starting policy which is issued from the (very noisy) DR optimization process. Interestingly, even with these very noisy policies, IWOCS still outperforms other algorithms on average across environments. However the variance of obtained scores is quite high. The columns of Table 23 should be compared with those of Table 1 in the main text of the paper, showing that variance in IWOCS' performance with variables DR initial policy is mostly due to the large variance in DR's initial policies.

## P    IMPACT STATEMENT

This paper presents work whose goal is to advance the field of reinforcement learning. It tackles generic mathematical and computational challenges, which might have potential societal and technological consequences, none of which we feel must be specifically highlighted here.

## Q    LIMITATIONS

The IWOCS algorithm assumes that all transition kernels in the uncertainty set $\mathcal{T}$ are known during training. In real-world applications, obtaining such detailed information is not always feasible. This

Table 14: Halfcheetah 2, worst parameters search for each iteration over 10 seeds.

| | $\psi_1^0$ | $\psi_1^1$ | $J_{T_1}^{\pi_0}$ | $J_{T_1}^{\pi_1}$ | $\psi_2^0$ | $\psi_2^1$ | $J_{T_2}^{\pi_1}$ | $J_{T_2}^{\pi_2}$ | $\psi_3^0$ | $\psi_3^1$ | $J_{T_3}^{\pi_2}$ | $J_{T_3}^{\pi_3}$ | $\psi_4^0$ | $\psi_4^1$ | $J_{T_4}^{\pi_3}$ |
|---|---|---|---|---|---|---|---|---|---|---|---|---|---|---|---|
| 0 | 3.61 | 0.79 | 1.57 | 5.28 | 3.61 | 0.1 | 3.02 | 7.23 | 3.61 | 0.1 | 3.02 | - | - | - | - |
| 1 | 3.61 | 0.79 | 1.57 | 6.93 | 3.61 | 0.1 | 3.02 | 7.23 | 3.61 | 0.1 | 3.02 | - | - | - | - |
| 2 | 3.61 | 0.79 | 1.57 | 7.82 | 3.61 | 0.1 | 3.01 | 6.32 | 3.61 | 0.1 | 3.01 | - | - | - | - |
| 3 | 3.61 | 0.79 | 1.57 | 5.70 | 3.61 | 0.1 | 3.09 | 6.40 | 3.61 | 0.1 | 3.09 | - | - | - | - |
| 4 | 3.61 | 0.79 | 1.57 | 7.83 | 3.61 | 0.1 | 3.02 | 7.26 | 3.61 | 0.1 | 3.02 | - | - | - | - |
| 5 | 3.61 | 0.79 | 1.57 | 4.29 | 3.61 | 0.1 | 3.02 | 7.04 | 3.61 | 0.1 | 3.02 | - | - | - | - |
| 6 | 3.61 | 0.79 | 1.57 | 4.80 | 3.61 | 0.1 | 2.98 | 7.62 | 3.61 | 0.1 | 2.98 | - | - | - | - |
| 7 | 3.61 | 0.79 | 1.57 | 7.70 | 3.61 | 0.1 | 3.03 | 8.56 | 3.61 | 0.1 | 3.03 | - | - | - | - |
| 8 | 3.61 | 0.79 | 1.57 | 5.17 | 3.61 | 0.1 | 3.02 | 6.28 | 3.61 | 0.1 | 3.02 | - | - | - | - |
| 9 | 3.61 | 0.79 | 1.57 | 4.89 | 3.61 | 0.1 | 3.03 | 6.79 | 3.61 | 0.1 | 3.03 | - | - | - | - |

Table 15: Hopper 2, worst parameters search for each iteration over 10 seeds.

| | $\psi_1^0$ | $\psi_1^1$ | $J_{T_1}^{\pi_0}$ | $J_{T_1}^{\pi_1}$ | $\psi_2^0$ | $\psi_2^1$ | $J_{T_2}^{\pi_1}$ | $J_{T_2}^{\pi_2}$ | $\psi_3^0$ | $\psi_3^1$ | $J_{T_3}^{\pi_2}$ | $J_{T_3}^{\pi_3}$ | $\psi_4^0$ | $\psi_4^1$ | $J_{T_4}^{\pi_3}$ |
|---|---|---|---|---|---|---|---|---|---|---|---|---|---|---|---|
| 0 | 2.13 | 0.1 | 3.35 | 2.43 | 2.13 | 0.1 | 3.35 | - | - | - | - | - | - | - | - |
| 1 | 2.13 | 0.1 | 3.35 | 1.07 | 2.13 | 0.1 | 3.35 | - | - | - | - | - | - | - | - |
| 2 | 2.13 | 0.1 | 3.35 | 3.51 | 2.13 | 0.1 | 3.35 | - | - | - | - | - | - | - | - |
| 3 | 2.13 | 0.1 | 3.35 | 1.35 | 2.13 | 0.1 | 3.35 | - | - | - | - | - | - | - | - |
| 4 | 2.13 | 0.1 | 3.35 | 3.32 | 2.13 | 0.1 | 3.35 | - | - | - | - | - | - | - | - |
| 5 | 2.13 | 0.1 | 3.35 | 3.53 | 2.13 | 0.1 | 3.35 | - | - | - | - | - | - | - | - |
| 6 | 2.13 | 0.1 | 3.35 | 1.20 | 2.13 | 0.1 | 3.35 | - | - | - | - | - | - | - | - |
| 7 | 2.13 | 0.1 | 3.35 | 3.33 | 2.13 | 0.1 | 3.35 | - | - | - | - | - | - | - | - |
| 8 | 2.13 | 0.1 | 3.35 | 3.48 | 2.13 | 0.1 | 3.35 | - | - | - | - | - | - | - | - |
| 9 | 2.13 | 0.1 | 3.35 | 1.90 | 2.13 | 0.1 | 3.35 | - | - | - | - | - | - | - | - |

Table 16: Walker 2, worst parameters search for each iteration over 10 seeds.

| | $\psi_1^0$ | $\psi_1^1$ | $J_{T_1}^{\pi_0}$ | $J_{T_1}^{\pi_1}$ | $\psi_2^0$ | $\psi_2^1$ | $J_{T_2}^{\pi_1}$ | $J_{T_2}^{\pi_2}$ | $\psi_3^0$ | $\psi_3^1$ | $J_{T_3}^{\pi_2}$ | $J_{T_3}^{\pi_3}$ | $\psi_4^0$ | $\psi_4^1$ | $J_{T_4}^{\pi_3}$ |
|---|---|---|---|---|---|---|---|---|---|---|---|---|---|---|---|
| 0 | 3.22 | 0.1 | 3.77 | 3.27 | 3.22 | 0.1 | 4.1 | 1.57 | 3.22 | 0.1 | 4.1 | - | - | - | - |
| 1 | 3.22 | 0.1 | 3.77 | 4.92 | 3.22 | 0.1 | 4.1 | 6.6 | 3.22 | 0.1 | 4.1 | - | - | - | - |
| 2 | 3.22 | 0.1 | 3.77 | 3.17 | 3.22 | 0.1 | 4.14 | 5.47 | 3.22 | 0.1 | 4.14 | - | - | - | - |
| 3 | 3.22 | 0.1 | 3.77 | 5.91 | 3.22 | 0.1 | 4.13 | 5.09 | 3.22 | 0.1 | 4.13 | - | - | - | - |
| 4 | 3.22 | 0.1 | 3.77 | 6.72 | 3.22 | 0.1 | 4.11 | 5.4 | 3.22 | 0.1 | 4.11 | - | - | - | - |
| 5 | 3.22 | 0.1 | 3.77 | 3.93 | 3.22 | 0.1 | 4.1 | 5.87 | 3.22 | 0.1 | 4.1 | - | - | - | - |
| 6 | 3.22 | 0.1 | 3.77 | 4.67 | 3.22 | 0.1 | 3.8 | 6.25 | 3.22 | 0.59 | 3.81 | 3.22 | 0.59 | 3.92 | 5.4 |
| 7 | 3.22 | 0.1 | 3.77 | 6.54 | 3.22 | 1.08 | 4.07 | 3.33 | 3.22 | 0.1 | 4.07 | - | - | - | - |
| 8 | 3.22 | 0.1 | 3.77 | 4.88 | 3.22 | 0.1 | 4.15 | 4.37 | 3.22 | 0.1 | 4.15 | - | - | - | - |
| 9 | 3.22 | 0.1 | 3.77 | 3.99 | 3.22 | 0.1 | 4.14 | 3.28 | 3.22 | 0.1 | 4.14 | - | - | - | - |

Table 17: Ant 3, worst parameters search for each iteration over 10 seeds.

| | $\psi_1^0$ | $\psi_1^1$ | $\psi_1^3$ | $J_{T_1}^{\pi_0}$ | $J_{T_1}^{\pi_1}$ | $\psi_2^0$ | $\psi_2^1$ | $\psi_2^3$ | $J_{T_2}^{\pi_1}$ | $J_{T_2}^{\pi_2}$ | $\psi_3^0$ | $\psi_3^1$ | $\psi_3^3$ | $J_{T_3}^{\pi_2}$ | $J_{T_3}^{\pi_3}$ | $\psi_4^0$ | $\psi_4^1$ | $\psi_4^3$ | $J_{T_4}^{\pi_3}$ |
|---|---|---|---|---|---|---|---|---|---|---|---|---|---|---|---|---|---|---|---|
| 0 | 0.68 | 0.01 | 0.309 | -1.21 | 3.59 | 0.68 | 0.01 | 0.309 | -1.21 | 4.59 | - | - | - | - | - | - | - | - | - |
| 1 | 0.68 | 0.01 | 0.309 | -1.21 | 0.54 | 1.26 | 1.505 | 2.701 | -1.08 | 2.54 | 1.26 | 1.804 | 2.402 | -0.9 | 2.3 | 1.26 | 1.804 | 2.402 | -0.9 |
| 2 | 0.68 | 0.01 | 0.309 | -1.21 | 5.92 | 2.13 | 0.309 | 1.804 | -0.44 | 3.92 | 2.13 | 0.309 | 1.804 | -0.44 | - | - | - | - | - |
| 3 | 0.68 | 0.01 | 0.309 | -1.21 | -1.35 | 2.13 | 0.309 | 1.804 | -0.44 | 1.23 | 2.13 | 0.309 | 1.804 | -0.44 | - | - | - | - | - |
| 4 | 0.68 | 0.01 | 0.309 | -1.21 | 2.58 | 0.68 | 0.608 | 2.701 | -1.02 | 2.71 | 0.1 | 0.309 | 2.402 | -0.75 | 2.4 | 0.1 | 0.309 | 2.402 | -0.75 |
| 5 | 0.68 | 0.01 | 0.309 | -1.21 | 3.07 | 2.42 | 0.608 | 2.103 | -0.78 | 5.29 | 2.42 | 0.608 | 2.103 | -0.78 | - | - | - | - | - |
| 6 | 0.68 | 0.01 | 0.309 | -1.21 | 2.07 | 2.42 | 1.505 | 0.01 | -0.92 | 3.33 | 2.71 | 0.309 | 2.103 | -0.63 | 5.42 | 2.71 | 0.309 | 2.103 | -0.63 |
| 7 | 0.68 | 0.01 | 0.309 | -1.21 | 1.94 | 0.39 | 1.804 | 0.608 | -1.02 | 2.21 | 0.1 | 1.505 | 2.103 | -0.84 | 6.08 | 0.1 | 1.505 | 2.103 | -0.84 |
| 8 | 0.68 | 0.01 | 0.309 | -1.21 | 2.33 | 2.13 | 0.309 | 1.804 | -0.44 | 4.21 | 2.13 | 0.309 | 1.804 | -0.43 | - | - | - | - | - |
| 9 | 0.68 | 0.01 | 0.309 | -1.21 | 1.44 | 2.13 | 0.309 | 1.804 | -0.44 | 2.1 | 2.13 | 0.309 | 1.804 | -0.45 | - | - | - | - | - |

Table 18: Halfcheetah , worst parameters search for each iteration over 10 seeds.

| | $\psi_1^0$ | $\psi_1^1$ | $\psi_1^3$ | $J_{T_1}^{\pi_0}$ | $J_{T_1}^{\pi_1}$ | $\psi_2^0$ | $\psi_2^1$ | $\psi_2^3$ | $J_{T_2}^{\pi_1}$ | $J_{T_2}^{\pi_2}$ | $\psi_3^0$ | $\psi_3^1$ | $\psi_3^3$ | $J_{T_3}^{\pi_2}$ | $J_{T_3}^{\pi_3}$ | $\psi_4^0$ | $\psi_4^1$ | $\psi_4^3$ | $J_{T_4}^{\pi_3}$ |
|---|---|---|---|---|---|---|---|---|---|---|---|---|---|---|---|---|---|---|---|
| 0 | 2.44 | 5.62 | 0.39 | 0.28 | 10.11 | 2.44 | 5.62 | 0.39 | 0.28 | - | - | - | - | - | - | - | - | - | - |
| 1 | 2.44 | 5.62 | 0.39 | 0.28 | 4.31 | 2.83 | 2.86 | 0.1 | 0.37 | 8.79 | 2.83 | 2.86 | 0.1 | 0.37 | - | - | - | - | - |
| 2 | 2.44 | 5.62 | 0.39 | 0.28 | 8.74 | 2.44 | 5.62 | 0.39 | 0.28 | - | - | - | - | - | - | - | - | - | - |
| 3 | 2.44 | 5.62 | 0.39 | 0.28 | 9.78 | 2.44 | 5.62 | 0.39 | 0.28 | - | - | - | - | - | - | - | - | - | - |
| 4 | 2.44 | 5.62 | 0.39 | 0.28 | 10.30 | 2.44 | 5.62 | 0.39 | 0.28 | - | - | - | - | - | - | - | - | - | - |
| 5 | 2.44 | 5.62 | 0.39 | 0.28 | 8.9 | 2.83 | 6.31 | 0.39 | 0.35 | 8.0 | 2.83 | 6.31 | 0.39 | 0.35 | - | - | - | - | - |
| 6 | 2.44 | 5.62 | 0.39 | 0.28 | 9.8 | 2.83 | 6.31 | 0.1 | 0.30 | 8.37 | 2.83 | 6.31 | 0.01 | 0.31 | 8.22 | 2.83 | 6.31 | 0.1 | 0.34 |
| 7 | 2.44 | 5.62 | 0.39 | 0.28 | 8.39 | 3.22 | 5.62 | 0.1 | 0.49 | 7.04 | 3.22 | 5.62 | 0.1 | 0.49 | - | - | - | - | - |
| 8 | 2.44 | 5.62 | 0.39 | 0.28 | 11.60 | 2.44 | 5.62 | 0.39 | 0.28 | - | - | - | - | - | - | - | - | - | - |
| 9 | 2.44 | 5.62 | 0.39 | 0.28 | 9.15 | 2.44 | 5.62 | 0.39 | 0.28 | - | - | - | - | - | - | - | - | - | - |

Table 19: Hopper 3, worst parameters search for each iteration over 10 seeds.

| | $\psi_1^0$ | $\psi_1^1$ | $\psi_1^3$ | $J_{T_1}^{\pi_0}$ | $J_{T_1}^{\pi_1}$ | $\psi_2^0$ | $\psi_2^1$ | $\psi_2^3$ | $J_{T_2}^{\pi_1}$ | $J_{T_2}^{\pi_2}$ | $\psi_3^0$ | $\psi_3^1$ | $\psi_3^3$ | $J_{T_3}^{\pi_2}$ | $J_{T_3}^{\pi_3}$ | $\psi_4^0$ | $\psi_4^1$ | $\psi_4^3$ | $J_{T_4}^{\pi_3}$ |
|---|---|---|---|---|---|---|---|---|---|---|---|---|---|---|---|---|---|---|---|
| 0 | 2.71 | 0.1 | 0.49 | 1.23 | 3.41 | 2.71 | 0.1 | 0.49 | 1.23 | 3.73 | - | - | - | - | - | - | - | - | - |
| 1 | 2.71 | 0.1 | 0.49 | 1.23 | 3.81 | 2.71 | 0.39 | 0.1 | 1.31 | 3.51 | 2.71 | 0.39 | 0.1 | 1.35 | 2.81 | 2.71 | 0.39 | 0.1 | 1.35 |
| 2 | 2.71 | 0.1 | 0.49 | 1.23 | 2.29 | 2.71 | 0.39 | 0.1 | 1.31 | 3.24 | 2.71 | 0.39 | 0.1 | 1.31 | - | - | - | - | - |
| 3 | 2.71 | 0.1 | 0.49 | 1.23 | 3.16 | 2.71 | 0.39 | 0.1 | 1.30 | 2.21 | 2.71 | 0.39 | 0.1 | 1.32 | 3.29 | 2.71 | 0.39 | 0.1 | 1.34 |
| 4 | 2.71 | 0.1 | 0.49 | 1.23 | 3.88 | 2.71 | 0.39 | 0.1 | 1.326 | 3.8 | 2.71 | 0.39 | 0.1 | 1.326 | 2.203 | 2.71 | 0.39 | 0.1 | 1.326 |
| 5 | 2.71 | 0.1 | 0.49 | 1.23 | 3.65 | 2.42 | 0.1 | 0.88 | 1.362 | 3.89 | 2.42 | 0.1 | 0.88 | 1.364 | 3.505 | 2.71 | 0.1 | 0.49 | 1.46 |
| 6 | 2.71 | 0.1 | 0.49 | 1.23 | 3.39 | 2.71 | 0.39 | 0.1 | 1.32 | 3.47 | 2.71 | 0.39 | 0.1 | 1.32 | - | - | - | - | - |
| 7 | 2.71 | 0.1 | 0.49 | 1.23 | 3.11 | 2.71 | 0.39 | 0.1 | 1.31 | 3.65 | 2.71 | 0.39 | 0.1 | 1.31 | - | - | - | - | - |
| 8 | 2.71 | 0.1 | 0.49 | 1.23 | 3.37 | 2.71 | 0.1 | 0.88 | 1.32 | 3.76 | 2.71 | 0.39 | 0.1 | 1.33 | 3.75 | 2.71 | 0.39 | 0.1 | 1.33 |
| 9 | 2.71 | 0.1 | 0.49 | 1.23 | 3.12 | 2.71 | 0.39 | 0.1 | 1.34 | 3.24 | 2.71 | 0.39 | 0.1 | 1.344 | 3.81 | 2.71 | 0.39 | 0.1 | 1.35 |

Table 20: Humanoid 3, worst parameters search for each iteration over 10 seeds.

| | $\psi_1^0$ | $\psi_1^1$ | $\psi_1^3$ | $J_{T_1}^{\pi_0}$ | $J_{T_1}^{\pi_1}$ | $\psi_2^0$ | $\psi_2^1$ | $\psi_2^3$ | $J_{T_2}^{\pi_1}$ | $J_{T_2}^{\pi_2}$ | $\psi_3^0$ | $\psi_3^1$ | $\psi_3^3$ | $J_{T_3}^{\pi_2}$ | $J_{T_3}^{\pi_3}$ | $\psi_4^0$ | $\psi_4^1$ | $\psi_4^3$ | $J_{T_4}^{\pi_3}$ |
|---|---|---|---|---|---|---|---|---|---|---|---|---|---|---|---|---|---|---|---|
| 0 | 14.41 | 0.59 | 0.1 | 6.88 | 15.37 | 14.41 | 0.59 | 0.1 | 6.88 | - | - | - | - | - | - | - | - | - | - |
| 1 | 14.41 | 0.59 | 0.1 | 6.88 | 8.19 | 14.41 | 0.59 | 0.1 | 6.88 | - | - | - | - | - | - | - | - | - | - |
| 2 | 14.41 | 0.59 | 0.1 | 6.88 | 14.05 | 12.82 | 1.57 | 0.1 | 8.41 | 12.02 | 12.82 | 1.57 | 0.1 | 8.41 | - | - | - | - | - |
| 3 | 14.41 | 0.59 | 0.1 | 6.88 | 14.18 | 14.41 | 0.59 | 0.1 | 7.41 | 8.23 | 14.41 | 0.59 | 0.1 | 7.41 | - | - | - | - | - |
| 4 | 14.41 | 0.59 | 0.1 | 6.88 | 13.42 | 14.41 | 0.59 | 0.1 | 6.88 | - | - | - | - | - | - | - | - | - | - |
| 5 | 14.41 | 0.59 | 0.1 | 6.88 | 10.43 | 14.41 | 0.59 | 0.1 | 6.88 | - | - | - | - | - | - | - | - | - | - |
| 6 | 14.41 | 0.59 | 0.1 | 6.88 | 11.38 | 14.41 | 0.59 | 0.1 | 7.10 | 13.39 | 14.41 | 0.59 | 0.1 | 7.10 | - | - | - | - | - |
| 7 | 14.41 | 0.59 | 0.1 | 6.88 | 12.78 | 14.41 | 0.59 | 0.1 | 7.10 | 11.86 | 14.41 | 0.59 | 0.1 | 7.10 | - | - | - | - | - |
| 8 | 14.41 | 0.59 | 0.1 | 6.88 | 13.20 | 14.41 | 2.06 | 0.1 | 7.171 | 9.27 | 14.41 | 2.06 | 0.1 | 7.171 | - | - | - | - | - |
| 9 | 14.41 | 0.59 | 0.1 | 6.88 | 15.43 | 14.41 | 2.06 | 0.1 | 7.175 | 14.37 | 14.41 | 2.06 | 0.1 | 7.175 | - | - | - | - | - |

Table 21: Walker 3, worst parameters search for each iteration over 10 seeds.

| | $\psi_1^0$ | $\psi_1^1$ | $\psi_1^3$ | $J_{T_1}^{\pi_0}$ | $J_{T_1}^{\pi_1}$ | $\psi_2^0$ | $\psi_2^1$ | $\psi_2^3$ | $J_{T_2}^{\pi_1}$ | $J_{T_2}^{\pi_2}$ | $\psi_3^0$ | $\psi_3^1$ | $\psi_3^3$ | $J_{T_3}^{\pi_2}$ | $J_{T_3}^{\pi_3}$ | $\psi_4^0$ | $\psi_4^1$ | $\psi_4^3$ | $J_{T_4}^{\pi_3}$ |
|---|---|---|---|---|---|---|---|---|---|---|---|---|---|---|---|---|---|---|---|
| 0 | 3.22 | 4.02 | 0.1 | 3.92 | 4.84 | 3.22 | 4.02 | 0.1 | 3.92 | - | - | - | - | - | - | - | - | - | - |
| 1 | 3.22 | 4.02 | 0.1 | 3.92 | 5.14 | 2.05 | 3.53 | 0.69 | 4.28 | 4.44 | 2.05 | 3.53 | 0.69 | 4.28 | - | - | - | - | - |
| 2 | 3.22 | 4.02 | 0.1 | 3.92 | 5.36 | 3.22 | 4.02 | 0.1 | 3.92 | - | - | - | - | - | - | - | - | - | - |
| 3 | 3.22 | 4.02 | 0.1 | 3.92 | 4.60 | 3.22 | 2.55 | 0.1 | 4.21 | 4.75 | 3.22 | 2.55 | 0.1 | 4.21 | - | - | - | - | - |
| 4 | 3.22 | 4.02 | 0.1 | 3.92 | 5.24 | 2.05 | 3.53 | 0.69 | 4.287 | 4.49 | 2.05 | 3.53 | 0.69 | 4.287 | - | - | - | - | - |
| 5 | 3.22 | 4.02 | 0.1 | 3.92 | 3.16 | 2.05 | 3.53 | 0.69 | 4.28 | 3.3 | 2.05 | 3.53 | 0.69 | 4.28 | - | - | - | - | - |
| 6 | 3.22 | 4.02 | 0.1 | 3.92 | 4.87 | 2.44 | 4.02 | 0.1 | 4.06 | 4.21 | 3.22 | 2.55 | 0.1 | 4.61 | 4.42 | 3.22 | 2.55 | 0.1 | 4.61 |
| 7 | 3.22 | 4.02 | 0.1 | 3.92 | 4.70 | 3.61 | 0.1 | 0.1 | 4.45 | 5.92 | 3.61 | 0.1 | 0.1 | 4.45 | - | - | - | - | - |
| 8 | 3.22 | 4.02 | 0.1 | 3.92 | 5.91 | 3.61 | 0.1 | 0.1 | 4.19 | 5.97 | 3.61 | 0.1 | 0.1 | 4.19 | - | - | - | - | - |
| 9 | 3.22 | 4.02 | 0.1 | 3.92 | 3.51 | 3.61 | 0.1 | 0.1 | 4.29 | 4.23 | 3.61 | 0.1 | 0.1 | 4.29 | - | - | - | - | - |

reliance on precise uncertainty set knowledge limits the practical usage of our algorithms in some cases.

Another limitation of the Deep IWOCS algorithm is the need for hand-tuning or grid-search for the threshold $\rho$. A promising approach is using a variance network to detect non-valid value functions automatically. We plan to work on this for future developments.

| Environment | $J_{T_1}^{\pi_0}$ |
|---|---|
| Ant 2 | $-0.43 \pm 0.96$ |
| Ant 3 | $-0.32 \pm 0.50$ |
| HalfCheetah 2 | $0.66 \pm 0.30$ |
| HalfCheetah 3 | $0.41 \pm 0.11$ |
| Hopper 2 | $1.78 \pm 2.16$ |
| Hopper 3 | $2.33 \pm 1.44$ |
| HumanoidStandup 2 | $0.51 \pm 0.50$ |
| HumanoidStandup 3 | $0.45 \pm 0.89$ |
| InvertedPendulum 2 | $2.32 \pm 0.42$ |
| Walker 2 | $1.06 \pm 0.29$ |
| Walker 3 | $1.74 \pm 0.40$ |
| Aggregated | $0.96 \pm 0.72$ |

Table 22: Worst-case score distribution of $\pi_0$ trained with DR on each environment.

Table 23: Avg. of normalized worst-case performance over 10 seed for IWOCS* and IWOCS without fixed initial policy

| Environment | IWOCS* without fixed $\pi_0$ | IWOCS without fixed $\pi_0$ |
|---|---|---|
| Ant 2 | $0.12 \pm 0.81$ | $-0.27 \pm 0.44$ |
| Ant 3 | $0.19 \pm 0.52$ | $0.43 \pm 0.68$ |
| HalfCheetah 2 | $0.72 \pm 0.31$ | $0.75 \pm 0.28$ |
| HalfCheetah 3 | $0.51 \pm 0.18$ | $0.56 \pm 0.15$ |
| Hopper 2 | $5.41 \pm 1.15$ | $6.34 \pm 0.11$ |
| Hopper 3 | $4.34 \pm 0.32$ | $4.64 \pm 0.16$ |
| HumanoidStandup 2 | $0.69 \pm 0.30$ | $0.98 \pm 0.25$ |
| HumanoidStandup 3 | $0.86 \pm 0.72$ | $1.12 \pm 0.21$ |
| InvertedPendulum 2 | $2.82 \pm 0.00$ | $2.82 \pm 0.00$ |
| Walker 2 | $1.14 \pm 0.30$ | $1.23 \pm 0.10$ |
| Walker 3 | $1.90 \pm 0.42$ | $2.10 \pm 0.50$ |
| Aggregated | $1.7 \pm 0.46$ | $1.88 \pm 0.26$ |

