# OpenReview forum: "Solving robust MDPs as a sequence of static RL problems"
_ICLR.cc/2026/Conference — Submitted to ICLR 2026_

### Official Review · Reviewer_C5Nh · 2025-10-20

**Soundness:** 3
**Presentation:** 2
**Contribution:** 2
**Rating:** 4
**Confidence:** 3

**Summary:**

This paper revisits robust Markov Decision Processes (MDPs) from the perspective of the static model of transition uncertainty, as opposed to the commonly used dynamic or two-player adversarial model. It argues that under stationary policies and sa-rectangular uncertainty sets, the two formulations are equivalent. Building on this insight, the authors propose the Incremental Worst-Case Search (IWOCS) meta-algorithm, which iteratively identifies worst-case transition models and solves a sequence of standard (non-robust) RL problems. The method decouples policy optimization from adversarial search and is implemented using value iteration and a deep RL version. Experiments on MuJoCo benchmarks show that IWOCS achieves competitive or superior worst- and average-case performance compared to existing robust RL methods.

**Strengths:**

1. The paper introduces a novel static-model framework for robust MDPs, providing new insights into environment uncertainty through the equivalence between static and dynamic formulations under stationary policies and rectangular uncertainty.

2. The proposed IWOCS framework is conceptually simple, modular, and highly scalable to both tabular and deep RL settings.

3. Extensive experiments on MuJoCo benchmarks demonstrate strong worst-case performance, confirming the practical effectiveness of IWOCS compared with state-of-the-art robust RL methods.

**Weaknesses:**

1. The paper’s writing quality is uneven, with inconsistent comma usage, incorrect citation formatting (e.g., line 305), and an empty Appendix A.

2. The related work section largely overlooks recent advances (past 3 years) in robust and distributionally robust RL.

3. The framework claims to decouple policy optimization from the adversary, yet if the agent picks transition models, it effectively remains a two-player adversarial process.

4. The choice of discrete uncertainty sets is not well-motivated. Most modern robust MDP studies consider continuous uncertainty set, which provides stronger theoretical guarantees and broader coverage.

5. Section 4 introduces a simpler process without explaining what is simplified, why it is necessary, or how it impacts theoretical soundness.

6. Algorithm 1 is poorly described: the value function computation is missing; $\mathcal{T}_ {i}$ is defined but never used; $T_i$ is not properly updated (the algorithm may stagnate); and the “find worst $T_{i+1}$” step is ambiguous.  These issues make the procedure difficult to reproduce.

7. Figure 1 lacks an x-axis label, and IWOCS appears to have only two plotted points, limiting interpretability.

8. Appendix F’s pseudocode introduces undefined variables (e.g., $T_{i+1}$) and inconsistent notation, rendering the algorithm incomplete.

**Questions:**

1. What is the motivation for using discrete uncertainty sets instead of continuous ones? How does this affect the optimality of IWOCS?

2. How does IWOCS fundamentally differ from two-player adversarial training, given that it still identifies worst-case transitions?

3. What exactly is meant by the simpler process mentioned in Section 4? What is simplified, and at what theoretical cost?

4. Is IWOCS trained online or offline, and how are samples collected across iterations?

5. Why were only three baselines (M2TD3, M3DDPG, RARL) selected? Have newer robust RL methods (2022–2025) been considered?

6. Could you specify the normalization formula used for Tables 1–2 and provide raw reward values for reproducibility?

7. Why does the Ant environment perform significantly worse than others?

---

### Official Review · Reviewer_GhBY · 2025-10-29

**Soundness:** 2
**Presentation:** 3
**Contribution:** 3
**Rating:** 4
**Confidence:** 3

**Summary:**

This paper reexamines robust RL through the lens of static robust MDPs. It demonstrates that a standard robust MDP can be solved by decomposing it into a sequence of static RL problems, replacing the standard min–max formulation with an iterative worst-case search. The proposed algorithm IWOCS alternates between standard RL in a fixed so-far worst environment and identifying the worst environment. Experiments on MuJoCo benchmarks (Ant, Hopper, and HalfCheetah) show that IWOCS achieves good robustness and stability compared to prior methods like RARL and M2TD3.

**Strengths:**

1. The main idea of transforming robust MDPs as a sequence of static RL problems is insightful with well established mathematical explanation.
2. The proposed IWOCS algorithm performs well across benchmarks, showing stronger robustness and solid average returns.
3. The paper is well written and easy to follow, with clear structure and good intuition.

**Weaknesses:**

1. The algorithm is computationally expensive. It requires storing multiple Q-functions (line 5 of algorithm 1) and solving several full RL problems, which limits scalability to large-scale or high-dimensional settings. The worst-environment search is also heuristic and unstable across tasks (IWOCS vs. IWOCS* show noticeable gaps among different tasks) with no guarantee.
2. There’s no analysis of sample complexity, or how sensitive the algorithm is if the worst env identification is imperfect (I feel this is not easy for the continuous action/state )

**Questions:**

1. Is there reliable algorithm for worst env search? if not, can you analyze the algorithm's convergence given the retrieved env is not the worst one?
2. Could you analyze the sample complexity of this algorithm?
3. Could you add SAC as one of the baseline?

---

### Official Review · Reviewer_RTSD · 2025-10-30

**Soundness:** 2
**Presentation:** 2
**Contribution:** 1
**Rating:** 2
**Confidence:** 4

**Summary:**

This paper introduced the IWOCS method that finds the optimal policy robust to a set of pre-defined environmental transitions $\mathcal{T}$. Specifically, it iteratively finds the transition $T_i$ that minimizes $V^{\pi_{i-1}}$ from the previous iteration as well as the policy $\pi_i$ that maximizes the pessimistic value from the all the $T_i$ and $V^{\pi_i}$ it has interacted with before. Note that the algorithm is operated based on naive sampling implicitly assuming that sufficient samples can be obtained such that both $T = argmin {V_T}^{\pi_{i-1}}$ can be achieved almost surely and $V^*_{T_i}$ can be estimated without error in each iteration.

**Strengths:**

* The motivations are clearly conveyed by the paper and the approach is straightforward.
* The method is tested over various enviroments and compared with some baselines.

**Weaknesses:**

* The main concern from the reviewer is that the method is implicitly dependent on the fact that in each iteration the values functions $Q*$ and $V*$ can be perfectly obtained, and that $T_i$ can be found to minimize $V_T^\pi$. This might be doable in relatively small and discrete environments where the transition set $\mathcal{T}$ is also discrete. A number of concerns were raised from here.
  * First, the $T_i$ in each iteration is found by using some evolution algorithms/strategies -- what is the optimality guarantee/error bound/regret there that each time $T_i$ minimizes $V^\pi$ globally (when $\mathcal{T}$ and environmental transitions are both continuous)? If $T_i$ could not be solved perfectly in each iteration, how would it affect the optimality of the policy?
  * What is the sample complexity of finding $T_i$?
  * In non-discrete environmental transitions, how ${Q*_T}$ are obtained? If intractable, assuming that $Q*_T$ can be estimated with some error. Then could it violate the monotonicity property (property 2)? If this property is violated, would the algorithm still work? Could the authors show some guarantee, or the conditions, that the monotonicity could still be preserved even if $Q*_T$ could not be perfectly estimated? Or the other way around, if the monotonicity is not strictly preserved, how would it affect the optimality of the policy?
  * Even if the questions above could not be justified theoretically, could the authors validate them through numerical simulations (maybe start with the toy example and potentially extending to more complexed continuous environments)?

* The method also requires the set $\mathcal{T}$ to be fully known *a priori*. So the scope of this work is arguable covered by most of the distributionally robust RL work [1-4 below as a non-exhaustive list], which can find a policy robust to *unknown* environmental disturbance. Two more concerns following this line.
  * These methods can be potentially directly applied to the problem setup considered in this paper. Moreover, these methods usually come with sample complexity analyses, convergence and/or regret/optimality guarantees.
  * The reviewer is also curious how they performs against IWOCS in the experimental setup considered in this paper?

[1] Ramesh, Shyam Sundhar, et al. "Distributionally robust model-based reinforcement learning with large state spaces." International Conference on Artificial Intelligence and Statistics. PMLR, 2024.

[2] Shi, Laixi, et al. "The curious price of distributional robustness in reinforcement learning with a generative model." Advances in Neural Information Processing Systems 36 (2023): 79903-79917.

[3] Liu, Zijian, et al. "Distributionally Robust $ Q $-Learning." International Conference on Machine Learning. PMLR, 2022.

[4] Tessler, Chen, Yonathan Efroni, and Shie Mannor. "Action robust reinforcement learning and applications in continuous control." International Conference on Machine Learning. PMLR, 2019.

**Questions:**

One additional minor comment/question

* Given that $\mathcal{T}$ is expected to be fully known -- should it still be called the "uncertainty set"? In the reviewer's opinion, naming them as "a set of environmental dynamics/transitions/parameters" seem to better aligned with how $\mathcal{T}$ was used in this work.

---

### Official Review · Reviewer_E1gk · 2025-10-30

**Soundness:** 3
**Presentation:** 3
**Contribution:** 3
**Rating:** 4
**Confidence:** 3

**Summary:**

This paper proposes a new meta-algorithm, Incremental Worst-Case Search (IWOCS), for solving robust Markov Decision Processes (MDPs) with transition function uncertainty. The authors focus on the static model of uncertainty, where the environment's transition dynamics are fixed for an entire episode, which is often more practical but harder to solve than the commonly used dynamic model where dynamics can change at every timestep. The IWOCS algorithm works by iteratively building a discrete set of worst-case transition models. This approach effectively decouples policy optimization (a standard RL problem) from the search for adversarial environments. Empirical results on MuJoCo benchmarks show that IWOCS is competitive with and often outperforms existing robust RL methods.

**Strengths:**

+ The paper's focus on static models is well-motivated. This model is a realistic representation of many real-world robustness problems.

+ The proposed method is validated with strong experimental results on the MuJoCo benchmarks.

+ The paper is well-structured and clearly written in the introduction and experiment sections.

**Weaknesses:**

-- Algorithmic Contributions are Unclear: It is not clear which method is the main proposed one. The grid-search-based IWOCS* outperforms the CMA-ES-based IWOCS in aggregate (Table 1). This suggests that either the more sophisticated CMA-ES is an ineffective or unnecessary component, or that the benchmark uncertainty spaces are not challenging enough to warrant it over a simple grid search.

-- Interpretation of Baseline Results: The interpretation of the baseline results is lacking. In both Table 1 (worst-case) and Table 2 (average), methods like M3DDPG and RARL show highly negative normalized scores, implying they perform worse than a non-robust vanilla TD3. This is a surprising and counter-intuitive result that requires a clear explanation.

-- Incomplete Related Work: The related work section is not comprehensive and appears to be missing citations to some recent and relevant work in robust RL (e.g., [1, 2]).

-- Disconnect Between Theory and Practice: There is a significant disconnect between the paper's theoretical motivation and its empirical validation. The core justification (static-dynamic equivalence, no-duality gap) relies on the sa-rectangularity assumption. However, the authors explicitly state their MuJoCo experiments do not respect the rectangularity assumption (Footnote 4).

#### [1] Reddi et al. “Robust Adversarial Reinforcement Learning via Bounded Rationality Curricula”, ICLR 2024
#### [2] dong et al. “Variational Adversarial Training Towards Policies with Improved Robustness”, AISTATS 2025

**Questions:**

- On Baseline Performance: Can you provide an interpretation for why established robust RL methods like M3DDPG and RARL perform significantly worse than the non-robust TD3 baseline in your experiments (Tables 1 and 2)?

- On Table 3: Please clarify the experimental setup for Table 3. It is mentioned that each line corresponds to a different random seed, then why the initialized values in the first 3 columns are the same.

- How do you see IWOCS scaling to high-dimensional uncertainty spaces where black-box optimization methods like CMA-ES are intractable?

---

### Meta-Review · Area_Chair_EHYJ · 2025-12-08

**Summary:**

**Summary of the Paper**: This paper proposes a new meta-algorithm, Incremental Worst-Case Search (IWOCS), for solving robust Markov Decision Processes (MDPs) with transition function uncertainty. The authors focus on the static model of uncertainty, where the environment's transition dynamics are fixed for an entire episode. The IWOCS algorithm works by iteratively building a discrete set of worst-case transition models. This approach effectively decouples policy optimization (a standard RL problem) from the search for adversarial environments. Empirical results on MuJoCo benchmarks show that IWOCS is competitive with and often outperforms existing robust RL methods.

**AC's overview**: While all the reviewers agreed that the proposed approach has some value, and the empirical results are valuable, there are several issues raised by the reviewers.

(1) There are already a lot of works on robust MDP frameworks, including uncertainty sets consisting of discrete transition models [A1]. It is not clear what the advantages of the proposed approach are compared to these robust MDP frameworks. The proposed approach did not compare with the existing robust MDP approaches.

(2) The proposed approach, while intuitive, lacks a convergence guarantee. Hence, it is unclear in practice whether the proposed approach will converge or not (or, under what conditions, they will converge).

(3) The reviewers also raised concerns about the empirical results. For example, the reviewer E1gk mentioned that `` The interpretation of the baseline results is lacking. In both Table 1 (worst-case) and Table 2 (average), methods like M3DDPG and RARL show highly negative normalized scores, implying they perform worse than a non-robust vanilla TD3. This is a surprising and counter-intuitive result that requires a clear explanation." The AC agrees with this evaluation. It is also not clear how  IWOCS scales to high-dimensional uncertainty spaces where black-box optimization methods like CMA-ES are intractable.

[A1]. Wang, Qiuhao, Chin Pang Ho, and Marek Petrik. "Policy gradient in robust mdps with global convergence guarantee." In International Conference on Machine Learning, pp. 35763-35797. PMLR, 2023.

**Reviewer Concerns:**

The authors did not provide any rebuttals. Hence, the concerns are unaddressed.

**Reviewer Scores:**

N/A

The authors did not provide any rebuttal.

---

### Decision · Program_Chairs · 2026-01-26

Reject